# Strain-regulated Gibbs free energy enables reversible redox chemistry of chalcogenides for sodium ion batteries

Minxia Jiang[1,3], Yingjie Hu[2,3], Baoguang Mao [1]✉, Yixin Wang[1], Zhen Yang[1], Tao Meng[1], Xin Wang[1] & Minhua Cao [1]✉

Manipulating the reversible redox chemistry of transition metal dichalcogenides for energy storage often faces great challenges as it is difficult to regulate the discharged products directly. Herein we report that tensile-strained $MoSe_2$ (TS-$MoSe_2$) can act as a host to transfer its strain to corresponding discharged product Mo, thus contributing to the regulation of Gibbs free energy change ($\Delta G$) and enabling a reversible sodium storage mechanism. The inherited strain results in lattice distortion of Mo, which adjusts the d-band center upshifted closer to the Fermi level to enhance the adsorbability of $Na_2Se$, thereby leading to a decreased $\Delta G$ of the redox chemistry between Mo/$Na_2Se$ and $MoSe_2$. Ex situ and in situ experiments revealed that, unlike the unstrained $MoSe_2$, TS-$MoSe_2$ shows a highly reversible sodium storage, along with an evidently improved reaction kinetics. This work sheds light on the study on electrochemical energy storage mechanism of other electrode materials.

Conversion-type transition metal dichalcogenides (TMDs), normally with a formula of $MX_2$ (M = Mo, V, W, Re; X = S, Se), are promising anode materials for lithium/sodium ion batteries owing to their high theoretical capacities[1–5]. The electrochemical storage mechanism research of these materials have attracted tremendous attention as it is the critical footstone for rational structure and morphology design of electrode materials to improve electrochemical performances[6–9]. Generally, the initial discharging process of $MX_2$ includes intercalation and conversion to form products M and $A_2X$ (A = Li, Na)[10]. In the following charging process, there are mainly two different pathways: $MX_2$ can be regenerated during the charging process following Eq. (1)[11], which means that the conversion reaction of $MX_2$ is reversible; or the formed $A_2X$ rather than M is oxidized to X upon the charging process, and eventually, $A_2X$/X becomes the main redox couple in the subsequent cycles according to Eq. (2), demonstrating the irreversibility of the conversion

reaction in $MX_2$[12]. Obviously,

$$M + 2A_2X \rightarrow MX_2 + 4A^+ + 4e^- \qquad (1)$$

$$A_2X \rightarrow X + 2A^+ + 2e^- \qquad (2)$$

taking $MoSe_2$ as an example, the essence of the sodium storage based on the irreversible mechanism actually has become a Na-Se battery, which will suffer from the shuttle effect of polyselenides and poor structural stability, thus leading to rapid capacity attenuation[13,14]. Therefore, manipulating $MX_2$ to follow a reversible reaction mechanism is highly necessary for achieving high electrochemical storage performance. When delving into the whole process in detail, it can be found that the key to the reversibility of the conversion reaction (Eq. 1)

---

[1]Key Laboratory of Cluster Science, Ministry of Education of China, Beijing Key Laboratory of Photoelectronic/Electrophotonic Conversion Materials, School of Chemistry and Chemical Engineering, Beijing Institute of Technology, Beijing 100081, P. R. China. [2]Nanjing Key Laboratory of Advanced Functional Materials, Nanjing Xiaozhuang University, Nanjing 211171, P. R. China. [3]These authors contributed equally: Minxia Jiang, Yingjie Hu. ✉e-mail: 7520200168@bit.edu.cn; caomh@bit.edu.cn

lies in whether Mo, the discharged product of MoSe$_2$, is capable of reacting with Na$_2$Se to re-form MoSe$_2$, because Mo is electrochemically inert and the Na$_2$Se has covalent characteristics[15,16]. Therefore, we assume that if we optimize the activity of the discharged products (Mo and Na$_2$Se) to drive the reaction between them, can we achieve the reversible conversion in MoSe$_2$? Nevertheless, this thinking is very challenging and has not yet been demonstrated in previous reports.

Recently, strain engineering in materials science has been deemed as an effective strategy to increase the intrinsic activity of the material by modifying its electronic properties, which is conductive to triggering the redox reaction[17–19]. For instance, Zhang et al. prepared tensile-strained Pd porous nanosheets and validated that the tensile strain could facilitate the conversion of Pd to PdO$_2$[20], while Chen et al. highlighted that the interfacial strain on reactants could promote the reaction by making the reaction energetically more favorable[21]. Specifically, in theory, the spontaneity of the redox reaction is determined by the change in Gibbs free energy of the process ($\Delta G$). In general, the smaller the $\Delta G$ between the products and the reactants, the more favorable the reaction is in thermodynamics. Moreover, the occurrence of the redox reaction also must conquer the kinetic energy barriers, which is closely associated with the $\Delta G$, as expressed by following Marcus equation[22]:

$$\Delta G^{\ddagger} = \frac{\lambda}{4}\left(1 + \frac{\Delta G}{\lambda}\right)^2 \qquad (3)$$

Where $\Delta G^{\ddagger}$ is the reaction activation energy and $\lambda$ is the reaction reorganization energy (usually between 0.5 – 1.0). Thus, the activation energy of the electron transfer reaction can be determined by the $\Delta G$. When $\Delta G > 0$, a smaller $\Delta G$ usually means a lower reaction activation energy barrier, which also implies accelerated reaction kinetics (Fig. 1a). Thus, it can be concluded that the effective modulation of the $\Delta G$ can not only improve the reaction thermodynamics but also promote the reaction kinetics.

Fortunately, it has been proved that, based on the quasi-harmonic density functional theory (DFT) calculations, the thermodynamically shifted Gibbs free energy ($G$) of redox reaction can be defined as a function of epitaxial strain ($\eta$) and temperature ($T$) according to the following relationship in Eq. (4)[23–25]:

$$\Delta G = \Delta E(\eta) + \Delta F(\eta, T) + \mu_M(T) \qquad (4)$$

Where $\mu_M$ is the chemical potential of the reactants, $\Delta E$ represents the zero-temperature contribution to the $G$, and $\Delta F$ represents the thermal contribution to the $G$. Based on this, to achieve a decreased value of $\Delta G$ and make the redox reaction energetically favorable, the key lies in applying lattice strain on the corresponding crystal (Fig. 1b). Therefore, it is highly expected that strain engineering may be an important enabler in the reversible transformation of sodiated MoSe$_2$ during the charging process by reducing $\Delta G$ value of the reaction. However, few research works have put efforts to understand the effect of strain engineering on the lithium/sodium

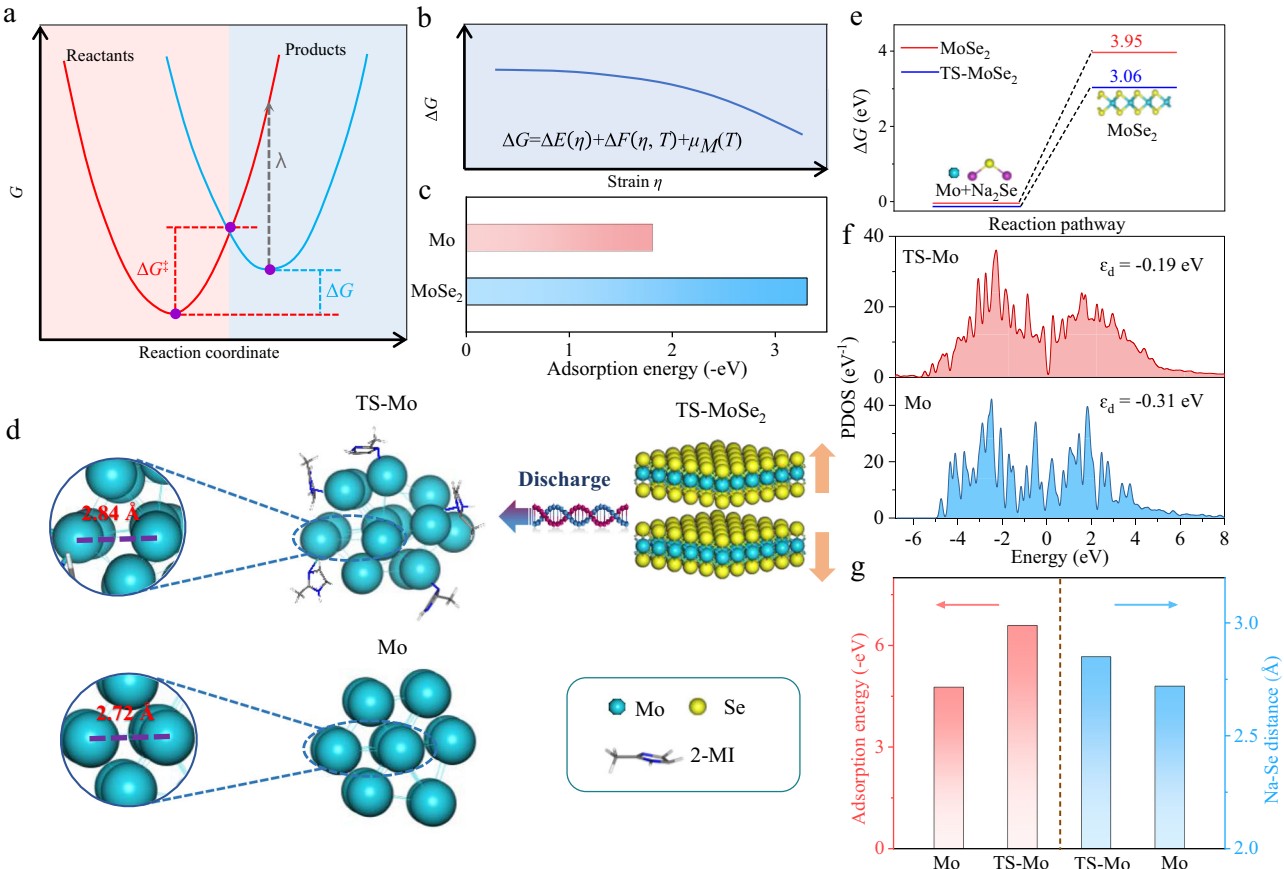

**Fig. 1 | Theoretical calculations. a** Schematic diagram of tuning reactants to make the redox reaction more energetically favorable. **b** The variation of $\Delta G$ associated with the redox reaction under epitaxial strain ($\eta$) at a constant temperature. **c** The adsorption energies per 2-MI molecule on MoSe$_2$ and Mo. **d** The atomic structure model of TS-Mo, Mo and the strain evolution from TS-MoSe$_2$ to TS-Mo. **e** The $\Delta G$ values per formula unit of the reaction between Na$_2$Se/Mo and MoSe$_2$ under strained and unstrained conditions. **f** PDOS of Mo-4d orbitals for TS-Mo and Mo. **g** The adsorption energies of per Na$_2$Se molecule on TS-Mo and Mo and the corresponding Na-Se distance evolution.

storage mechanism of conversion-type TMDs, owing to the lack of an efficient regulation methodology for the discharged products.

Herein, we demonstrate that tensile-strained $MoSe_2$ (denoted as TS-$MoSe_2$) can pass on the strain to its discharged product Mo (denoted as TS-Mo) by using 2-methylimidazole (2-MI) as a scaffold and rationalize the effect of strain engineering on its sodium storage mechanism. Both theoretical calculation and experimental results revealed that the tensile strain could activate inert Mo and $Na_2Se$ as well as reduce the $\Delta G$ value of their reaction. As a consequence, TS-$MoSe_2$ displays a highly reversible conversion mechanism of sodium storage, while its counterpart (unstrained $MoSe_2$) cannot be recovered during the charging process. Impressively, TS-$MoSe_2$ achieves high rate capacities and excellent cycling stability over a wide temperature range. This work provides a direction for the research on the alkali-metal-ion energy storage of conversion-type TMDs, which is essential to the rational design of high-performance electrode materials.

## Results

### Theoretical predictions for the strain effect on the activity of Mo

The feasibility of a strain engineering strategy that affects the reversibility of the sodium storage mechanism of $MoSe_2$ is assessed first by DFT calculations. It is worth mentioning that in our studies, 2-MI was chosen to exert the strain on the studied objects (Mo and $MoSe_2$) by means of its strong ligand effect[26], which can be further confirmed by its negative adsorption energies with Mo and $MoSe_2$, respectively (Fig. 1c and Supplementary Fig. 1). The atomic structure model of TS-Mo was constructed by introducing 2-MI species into Mo crystal structure (here, the 2-MI is reduced in size to make the underlying Mo structure more visible), which exerts the strain on Mo through the interaction between Mo and N atoms (Fig. 1d). It has been well addressed that the strong adsorption of the ligand will induce tensile strain on the adsorption sites[27,28], which significantly deform the structure to some extent and thereby might change the surface energy. Therefore, we first calculated the $\Delta G$ value of the studied reaction and the results clearly show that the reaction of TS-Mo and $Na_2Se$ to generate $MoSe_2$ (TS-Mo + $2Na_2Se \rightarrow MoSe_2 + 4Na^+ + 4e^-$) delivers a $\Delta G$ value of 3.06 eV (Fig. 1e, Supplementary Fig. 2, and Supplementary Table 1), which is much smaller than that (3.95 eV) of the same reversible reaction based on unstrained Mo. Furthermore, for unstrained Mo, this $\Delta G$ value corresponding to the reversible reaction is higher than that of the irreversible reaction (Mo + $Na_2Se \rightarrow$ Mo + Se + $2Na^+ + 2e^-$) (Supplementary Fig. 3), indicating that the irreversible reaction may occur preferentially than the reversible one for unstrained Mo. According to the Marcus equation in Fig. 1a, the smaller $\Delta G$ values also mean a reduced reaction energy barrier, implying that strain engineering could evidently promote the reaction not only in thermodynamics but also in kinetics. Furthermore, TS-Mo also exhibits an enlarged Mo-Mo bond length after employing the tensile strain (Fig. 1d), owing to the electronegativity difference between Mo and N atoms. The increased distance weakens the atomic interaction between Mo atoms, which is conductive to improve its reactivity. Therefore, the electronic structure of TS-Mo was also evaluated by partially density of states (PDOS), as shown in Fig. 1f. Obviously, TS-Mo shows an upshift of the d-band center toward the Fermi Level compared to unstrained Mo. Consequently, TS-Mo achieves a more negative adsorption energy value (−6.58 eV) than unstrained Mo (−4.77 eV) when interacting with $Na_2Se$, further providing a positive effect on the reaction between Mo and $Na_2Se$, which can be further confirmed by an enlarged Na-Se distance ($d_{Na-Se}$) of $Na_2Se$ adsorbed on TS-Mo (Fig. 1g and Supplementary Fig. 4). Taken together, it can be inferred that the strain engineering may show great potential in promoting the electrochemical reaction of Mo and $Na_2Se$ in energy storage devices. In our studies, Mo is the discharged product of $MoSe_2$, and therefore, in order to obtain TS-Mo experimentally, we can only start from the

$MoSe_2$ and also expect that TS-$MoSe_2$ can pass on its strain to Mo during the discharging process, thus possibly leading to the formation of TS-Mo.

### Materials synthesis and characterizations

Based on the above theoretical calculations, we then prepared TS-$MoSe_2$ by selenizing a Mo-precursor containing Mo source and 2-MI (Supplementary Fig. 5). Electron microscopy images clearly show that the resultant sample is featured with a hollow spherical structure with an average diameter of 150 nm (Fig. 2a and Supplementary Fig. 6), which inherits the spherical shape of the Mo-precursor (Supplementary Fig. 5d). The formation of the hollow structure follows the Kirkendall effect monitored by the time-dependent experiments (Supplementary Fig. 7). The specific surface area of the hollow TS-$MoSe_2$ was determined to be 17.68 $m^2 g^{-1}$ according to the Brunauer–Emmett–Teller method, which is slightly higher than that of the unstrained $MoSe_2$ (Supplementary Fig. 8). Additionally, further high-resolution transmission electron microscopy (HR-TEM) images display the typical features of few-layered $MoSe_2$ and the interlayer spacing is approximately 0.66 nm (Fig. 2b, c), slightly larger than the intrinsic (002) plane value[11,29]. Furthermore, the energy-dispersive spectroscopy (EDS) elemental mappings and line-scan profiles reveal that besides the Mo and Se elements that are uniformly distributed throughout the structure (Fig. 2d and Supplementary Fig. 9), N and C elements were also detected and that this sample also shows more mass loss (Supplementary Fig. 10). All of these results indirectly indicate that there may be some 2-MI species in $MoSe_2$. To further confirm this deduction, Fourier-transform infrared spectroscopy (FT-IR) was conducted. As shown in Supplementary Fig. 11, the bands at 1660, 1519, 1186, and 1030 $cm^{-1}$ are attributed to the stretching vibration of N-H, the skeleton vibration of the imidazole ring, and the stretching vibrations of C-N and C=C bonds, respectively, while the band at 837 $cm^{-1}$ is assigned to the vibration of C-H, proving the existence of the 2-MI species[30]. And the content of the 2-MI was determined to be about 4.10 wt% by CHN elemental analysis (Supplementary Table 2). Besides, X-ray photoelectron spectroscopy (XPS) measurements were performed to further study its existing form in $MoSe_2$ (Supplementary Fig. 12). As shown in Supplementary Fig. 12a, the survey XPS spectrum also shows the presence of the C and N elements in good agreement with the previous EDS results. The high-resolution N 1 s XPS spectrum (Supplementary Fig. 12b) shows the presence of pyridinic-N (397.6 eV), pyrrolic-N (399.7 eV), and graphitic-N (401.5 eV) and the peak at 397.6 eV can be assigned to N-Mo bond[31], suggesting the formation of coordination bond between 2-MI and $MoSe_2$. This interaction may lead to the tensile strain between the $MoSe_2$ layers, thus causing an expanded (002) interlayer spacing (Fig. 2c).

To gain further information on the tensile strain in the as-prepared TS-$MoSe_2$, a series of characterization techniques were used, such as X-ray diffraction (XRD), Raman spectrum, Fourier-transformed extended X-ray absorption fine structure (EXAFS) and XPS. As shown in Fig. 2e, the XRD pattern of TS-$MoSe_2$ is well consistent with the standard card of $MoSe_2$ (JCPDS card No. 29-0914) without any other crystalline impurities. However, compared to bulk $MoSe_2$, it is worth noting that the (002) diffraction peak (c-axis) of the as-prepared sample moves to a lower angle while the (100) peak shifts to a higher angle, indicating that there is a lattice expansion along the (002) direction and in-plane compression in TS-$MoSe_2$[32]. More importantly, the quantitative insight of the strain can be obtained by the XRD pattern according to the following Eq. (5)[33]:

$$strain\% = \frac{d(TS-MoSe_2) - d(MoSe_2)}{d(MoSe_2)} \times 100\% \qquad (5)$$

where $d$ represents the spacing of the corresponding planes, which can be deduced by the Bragg equation. Consequently, TS-$MoSe_2$ shows a

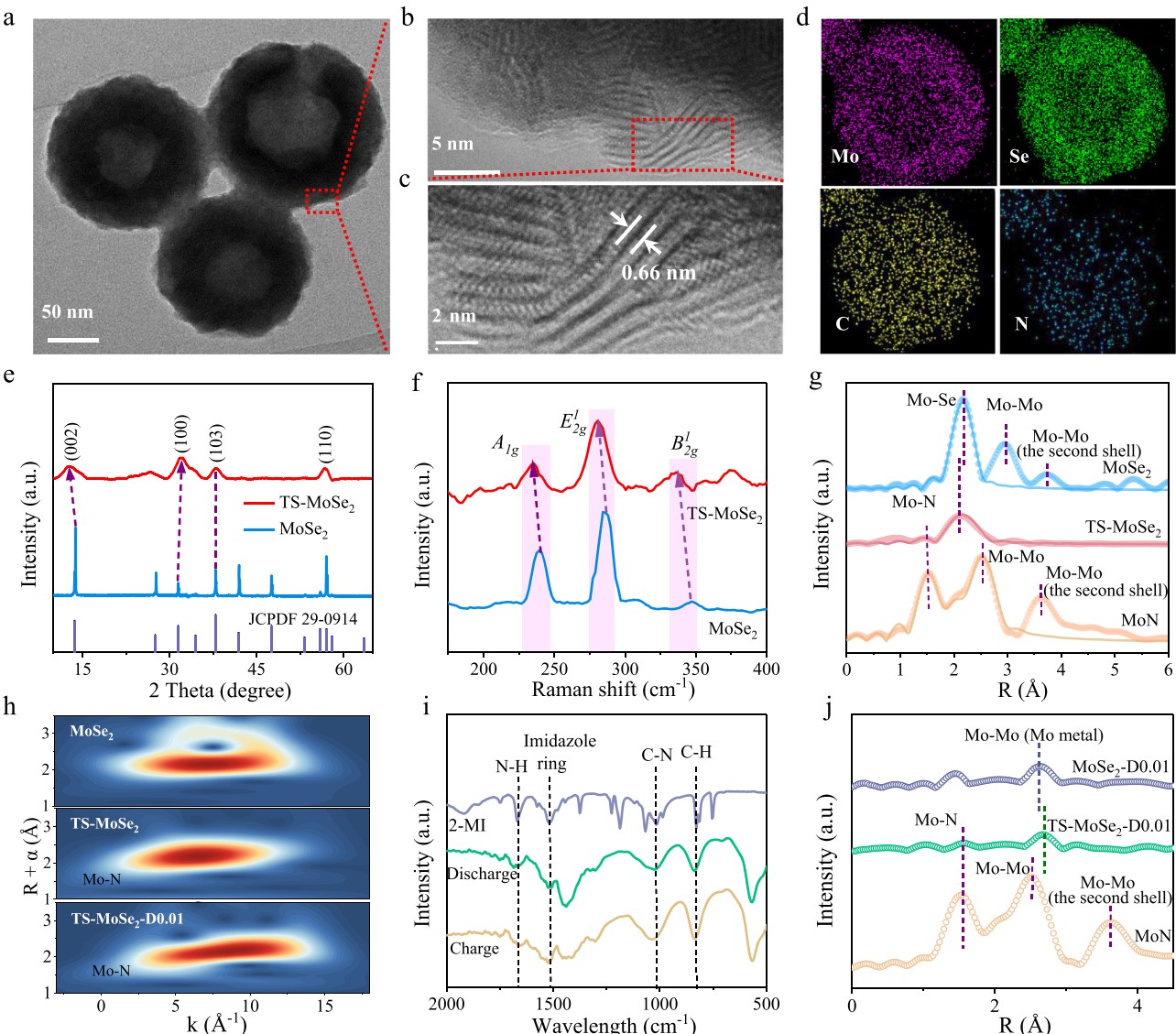

**Fig. 2 | Materials synthesis and characterizations. a** TEM, **b, c** HR-TEM images, and **d** elemental mapping images of TS-MoSe$_2$. **e** XRD patterns, **f** Raman spectra, and **g** The normalized Mo K-edge EXAFS spectra (circle) of TS-MoSe$_2$ and MoSe$_2$ as well as the corresponding EXAFS fitting curves (line). **h** Wavelet transform (WT) contour plots of MoSe$_2$, TS-MoSe$_2$, and MoN. **i** FT-IR spectra of 2-MI and the fully discharged and charged products of TS-MoSe$_2$. **j** The normalized Mo K-edge EXAFS spectra of MoN and the fully discharged product of TS-MoSe$_2$ (TS-MoSe$_2$-D0.01) and MoSe$_2$ (MoSe$_2$-D0.01).

tensile strain of about 6.34% along the c-axis and thereby an in-plane compressive strain of 3.15%. Furthermore, the distinct red shifts of $E^1_{2g}$, $A_{1g}$, and $B^1_{2g}$ in Raman spectrum of TS-MoSe$_2$ also suggest the existence of the strain (Fig. 2f), as the tensile strain leads to an expanded interlayer spacing, which can weaken the interlayer interaction force and in turn decrease the frequencies of $E^1_{2g}$, $A_{1g}$, and $B^1_{2g}$ vibration modes[32,34]. EXAFS spectroscopy in Fig. 2g displays the bond length evolution in TS-MoSe$_2$ and the shortened Mo-Se bond is clearly observed, which results from the in-plane compressive strain[35]. Besides, the curve fitting against the Fourier transforms of the EXAFS data for TS-MoSe$_2$ (Supplementary Fig. 13 and Supplementary Table 3) further proves the coordination bond of Mo and N and the corresponding coordination number is 0.8. Additionally, the wavelet transform (WT) plot of the Mo K-edge EXAFS for TS-MoSe$_2$ also presents a peak at 1.5 Å (Fig. 2h), which can be attributed to the dominance of the Mo-N scattering[36]. Meanwhile, the binding energies of Mo 3$d_{5/2}$ and Mo 3$d_{3/2}$ in the high-resolution Mo 3$d$ XPS spectrum of TS-MoSe$_2$ both shift to a low-energy side, suggesting that the strain may lead to the formation of 1T-MoSe$_2$ (Supplementary Fig. 14)[37]. The

above results indicate that the as-prepared TS-MoSe$_2$ does exist the out-plane tensile strain and in-plane compressive strain. In addition, we also studied discharged and charged TS-MoSe$_2$ by FT-IR (Fig. 2i) and found that the 2-MI species still exists, consistent with the result from Supplementary Fig. 11. Furthermore, the stability of the 2-MI molecule was further confirmed by linear sweep voltammetry (LSV) curves, in which no visible reduction peak ascribed to the 2-MI molecule is observed within the operating voltage window (Supplementary Fig. 15). Undoubtedly, the 2-MI species will continue to coordinate with the discharged product of TS-MoSe$_2$ (i.e., Mo), as confirmed by the Mo-N bond in the EXAFS and XPS spectra (Fig. 2j and Supplementary Fig. 16). It is noticeable that the coordination effect may enable Mo also with the tensile strain, that is, the strain in TS-MoSe$_2$ has been transferred to its discharged product. As depicted in Fig. 2j, TS-MoSe$_2$-D0.01 displays a peak at -2.69 Å, corresponding to the Mo-Mo bond in metallic Mo, which is slightly longer than that of the counterpart of unstrained MoSe$_2$, implying the existence of the tensile strain in metallic Mo induced by the coordination between 2-MI and Mo.

## Investigation of reversible sodium storage mechanism in TS-MoSe₂

Inspired by the positive influence that strain engineering has achieved on the redox reaction by the DFT calculations, we first performed ex situ XPS measurements to investigate the effect of the tensile strain on the sodium storage process. During the whole evolution process of discharging and charging, ten voltages were selected to evaluate the structural transformation of the TS-MoSe$_2$ electrode. As shown in the Mo 3$d$ XPS spectra (Fig. 3a), at the beginning of the discharging process (1.8 and 1.5 V), two main characteristic peaks at 228.83 and 231.93 eV that are related to 3$d_{5/2}$ and 3$d_{3/2}$ of Mo$^{4+}$ in MoSe$_2$ slightly shift towards the low binding energy, indicating the formation of the Na$_x$MoSe$_2$ intermediate. With further discharging (1.0 and 0.4 V), a component with lower binding energies at 227.43 (Mo 3$d_{5/2}$) and 230.53 eV (Mo 3$d_{3/2}$) appears and it can be assigned to metallic Mo[38], suggesting that the Na$_x$MoSe$_2$ has partly transformed into metallic Mo. At a fully discharged state, the Na$_x$MoSe$_2$ completely disappears and only metallic Mo is detected. Correspondingly, the Se 3$d$ peak at 54.5 eV first shifts to higher binding energy, and then restores to the original position, manifesting that Na$_2$Se finally forms through the polyselenide Na$_2$(Se)$_{1+n}$ ($n > 1$) during the discharging process (Fig. 3c)[39]. Afterward, in the following charging process, the peaks of both Mo 3$d$ and Se 3$d$ core levels can be fully recovered to their pristine state for TS-MoSe$_2$, and in contrast, for unstrained MoSe$_2$, metallic Mo is always present, and meanwhile, the elemental Se is eventually generated (Supplementary Figs. 17a, 18a). These changes can be observed more visually in corresponding 2D mapping images of the Mo 3$d$ and Se 3$d$ XPS spectra (Fig. 3b and Supplementary Figs. 17b, 18b, 19), which demonstrate that the strain engineering enables TS-MoSe$_2$ to follow highly reversible sodium storage mechanism in the discharging and charging processes.

The reversible sodium storage of TS-MoSe$_2$ was further confirmed by in situ Raman (Fig. 3d). As shown in Fig. 3e, TS-MoSe$_2$ exhibits a

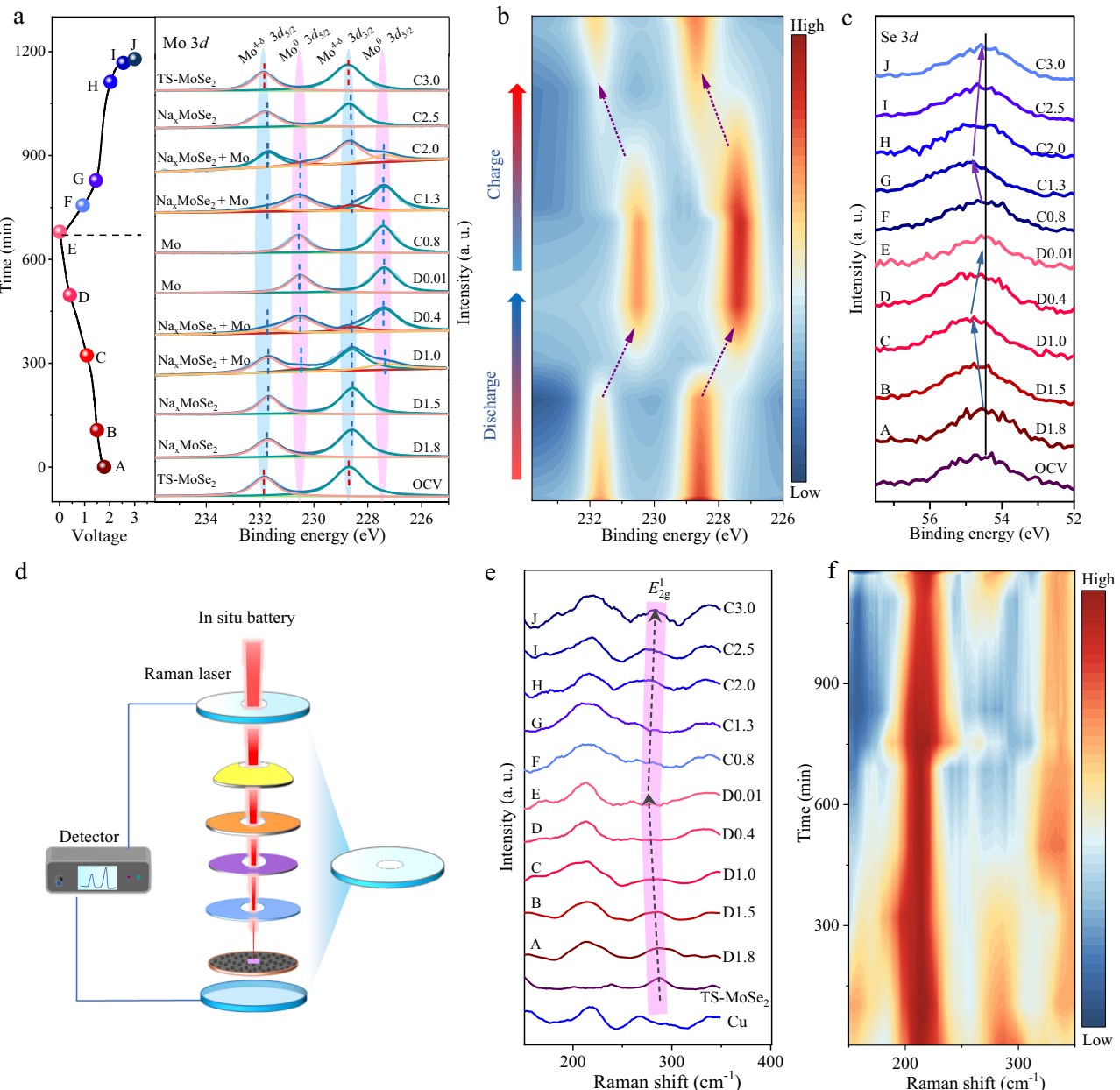

**Fig. 3 | Study on discharging and charging processes based on ex situ XPS and in situ Raman spectra. a–c** ex situ Mo 3$d$ XPS spectra (**a**) and corresponding mapping image(**b**), as well as Se 3$d$ XPS spectra (**c**) of TS-MoSe$_2$ during the initial discharging and charging processes. **d** Schematic illustration of in situ Raman measurement. **e, f** In situ Raman spectra (**e**) and corresponding mapping image (**f**) of TS-MoSe$_2$ during the initial discharging and charging processes.

prominent peak at about 285 cm$^{-1}$, corresponding to the $E^1_{2g}$ vibration mode of MoSe$_2$, and the peaks at 216 and 342 cm$^{-1}$ belong to the Cu foil-derived oxide[40]. During the discharging process, the $E^1_{2g}$ peak of MoSe$_2$ shows a slight red shift along with a decrease of peak intensity, which may come from the lattice expansion and disorder increase of TS-MoSe$_2$ induced by the intercalation of sodium ions. As the discharging process proceeds, the peak at 285 cm$^{-1}$ disappears completely, indicating that MoSe$_2$ is fully reduced. Reversibly, in the subsequent charging process, it is observed that the $E^1_{2g}$ peak of MoSe$_2$ is recovered again, indicating the regeneration of MoSe$_2$. The phenomenon is also proved by the peak color change in the mapping image in Fig. 3f. Furthermore, to exclude the influence of testing errors, we repeated the in situ Raman testing and the experimental results are basically consistent (Supplementary Fig. 20).

To further verify the above results, ex situ Mo K-edge X-ray adsorption spectroscopy (XAS) of TS-MoSe$_2$ was performed to track its valence state change and local atomic structure evolution during the electrochemical cycling. As depicted in Fig. 4a and Supplementary Fig. 21, during the discharging process, the absorption edge of Mo K-edge X-ray absorption near-edge structure (XANES) gradually shifts to a lower energy direction along with the insertion of sodium ions, manifesting that the valence state of Mo gradually decreases, namely, the reduction of MoSe$_2$ to Mo. After that, the absorption edge returns to the higher energy state until it almost coincides with the absorption edge of the pristine TS-MoSe$_2$ in the charging state (Fig. 4b and Supplementary Fig. 22). In addition, the corresponding wiggle/oscillatory features of the post-edge region of the pristine TS-MoSe$_2$, fully discharged TS-MoSe$_2$ (D0.01), fully charged TS-MoSe$_2$ (C3.0), and Mo foil can also reflect the variation in the local structure of TS-MoSe$_2$ during the electrochemical process. (Fig. 4c). Upon being fully discharged to 0.01 V, the appearance of the fingerprint feature of Mo foil at 20013.2, 20040.3, and 20084.1 eV supports the formation of metallic Mo[16]. In contrast, after the full charging, the aforesaid peaks almost recover to the original state of TS-MoSe$_2$, while the Mo foil-related features disappear, indicating that the discharging and charging processes of TS-MoSe$_2$ during the initial cycle are nearly fully reversible. It should be noted that the Mo K-edge XANES spectra of formed metallic Mo and the regenerated MoSe$_2$ are slightly different from those of corresponding Mo foil and pristine TS-MoSe$_2$, which may be caused by the ligand effect of imidazole and amorphous nature, respectively[41,42]. A similar change trend is also observed in the Se K-edge XANES (Supplementary Figs. 23, 24a–d). Specifically, during the initial discharging process, there are two obvious peaks located at 12661.08 and 12668.2 eV in the XANES spectra of TS-MoSe$_2$, which can be assigned to MoSe$_2$[43]. Nevertheless, these two peaks disappear and a peak appears at 12666.5 V upon discharging to 0.01 V, which corresponds to the generation of the discharged product Na$_2$Se. During the subsequent charging process, these peaks return to the original state, further indicating that the conversion reaction shows good reversibility.

Furthermore, the EXAFS spectra were applied to reveal the local structural changes of TS-MoSe$_2$ during the initial discharging and charging processes. As shown in Fig. 4d, the Mo K-edge EXAFS spectra of the pristine TS-MoSe$_2$ exhibit two obvious peaks at 2.11 and 3.09 Å, corresponding to Mo-Se interaction in the first coordination shell and Mo−Mo interaction, respectively[44,45]. With the intercalation of sodium ions, a peak appears at 2.69 Å accompanied by an increase of peak intensity, corresponding to the Mo−Mo bond in metallic Mo, which further confirms the generation of Mo during the discharging process[6]. The concentration changes of the TS-MoSe$_2$ and its discharged product Mo can be monitored by tracking the intensity changes of the corresponding peaks (Fig. 4e)[46]. Obviously, during the discharging process, the Mo-Se peak that belongs to TS-MoSe$_2$ gradually decreases in intensity, while the Mo-Mo peak (metallic Mo) continues to increase. Similarly, the corresponding Se K-edge EXAFS spectra (Supplementary Fig. 24e) also witnessed the gradual transformation of the Se-Mo bond

(MoSe$_2$) to the Se-Na bond (Na$_2$Se) upon discharging. In the subsequent charging process, the Mo-Mo (metallic Mo) and Se-Na peaks gradually disappear, while Mo-Se/Se-Mo and Se-Se (TS-MoSe$_2$) peaks become stronger. These observations further prove the excellent electrochemical reversibility of TS-MoSe$_2$. In addition, the Mo and Se K-edge XANES and EXAFS spectra of TS-MoSe$_2$ at D0.01 and C3.0 during the second and fifth cycles were also recorded, further confirming the reversible conversion of TS-MoSe$_2$ in the subsequent cycles (Fig. 4f and Supplementary Figs. 25–27). The whole sodium storage process of TS-MoSe$_2$ is illustrated in Supplementary Fig. 28, which goes through an intercalation and conversion reaction during the charging process and then the generated Mo and Na$_2$Se are reversibly converted into MoSe$_2$. By contrast, the unstrained MoSe$_2$ exhibits a different conversion mechanism compared with the TS-MoSe$_2$, as disclosed by its Mo K-edge XANES and EXAFS spectra. As shown in Fig. 4g and Supplementary Fig. 29, after charging at C3.0, the corresponding Mo-Mo (metallic Mo) peak at 20013.2 eV always exists, indicating that resultant metallic Mo did not participate in the subsequent reaction. In other words, the discharged product (metallic Mo) of MoSe$_2$ cannot regenerate MoSe$_2$ again in the charging process, and its sodium storage mechanism is irreversible. Therefore, combining ex situ XPS and XAS with in situ Raman spectra, it can be concluded that the strong interaction between the ligand and metal surface induces surface strain and subsequent surface reconstruction[47,48], which plays a significant role in the activation of Mo and thereby promotes the reversible sodium storage of MoSe$_2$, that is, TS-MoSe$_2$ exhibits a reversible sodium storage mechanism following Eq. (1), while unstrained MoSe$_2$ is irreversible as shown in Eq. (2).

## Electrochemical performance

In general, the reversible structure evolution will contribute to the improvement of battery performance. Thus, the electrochemical performances of TS-MoSe$_2$ and MoSe$_2$ as the anodes for SIBs were evaluated. Figure 5a shows cyclic voltammogram (CV) profiles of TS-MoSe$_2$ for the first four cycles within a potential range (V vs. Na/Na$^+$) of 0.01−3.0 V. The first cathodic scan presents two pronounced peaks at around 1.32 and 0.56 V, respectively, which are attributed to the intercalation of Na$^+$ into the MoSe$_2$ and the conversion reaction to form Na$_2$Se and metallic Mo nanograins. Meanwhile, the broad reduction peak ranging from 0.01 to 0.5 V in the first cycle is related to the electrolyte decomposition along with the formation of solid electrolyte interface (SEI) film[49,50]. During the following anodic scan, the distinct peak at 1.75 V, accompanied by a shoulder at 2.15 V, is ascribed to the conversion reaction between Mo and Na$_2$Se to form MoSe$_2$. These CV profiles overlap very well after the initial cycle, indicating the admirable reversibility and cyclic stability during the cycling process. By contrast, MoSe$_2$ (Supplementary Fig. 30) shows different peak positions in its CV curve of the first cathodic sweep. The absence of the peak located at 1.32 V suggests that the intercalation reaction hardly occurs in the initial cathodic process, which could be attributed to its relatively small interlayer distance and larger diffusion energy barrier[51]. Figure 5b shows their capacity vs. voltage (dQ/dV vs. V) plots at different selected cycles, where the redox peaks of TS-MoSe$_2$ corresponding to the reversible intercalation and conversion reaction hardly change in intensity even after 100 cycles. However, in sharp contrast, the redox peaks of the unstrained MoSe$_2$ electrode almost disappear after 100 cycles, which may ascribe to the loss of active materials owing to the shuttling of polyselenides[52]. Furthermore, the representative galvanostatic charge and discharge voltage curves of the TS-MoSe$_2$ and MoSe$_2$ anode in Supplementary Fig. 31 agree well with the above CV and dQ/dV results. Figure 5c and Supplementary Fig. 32 show the cycling performance of TS-MoSe$_2$ at the current density of 0.1 A g$^{-1}$. TS-MoSe$_2$ still maintains a specific capacity of 610 mA h g$^{-1}$ and an areal capacity of 0.36 mA h cm$^{-2}$ after 100 cycles, much higher than that of the MoSe$_2$ counterpart (350 mA h g$^{-1}$,

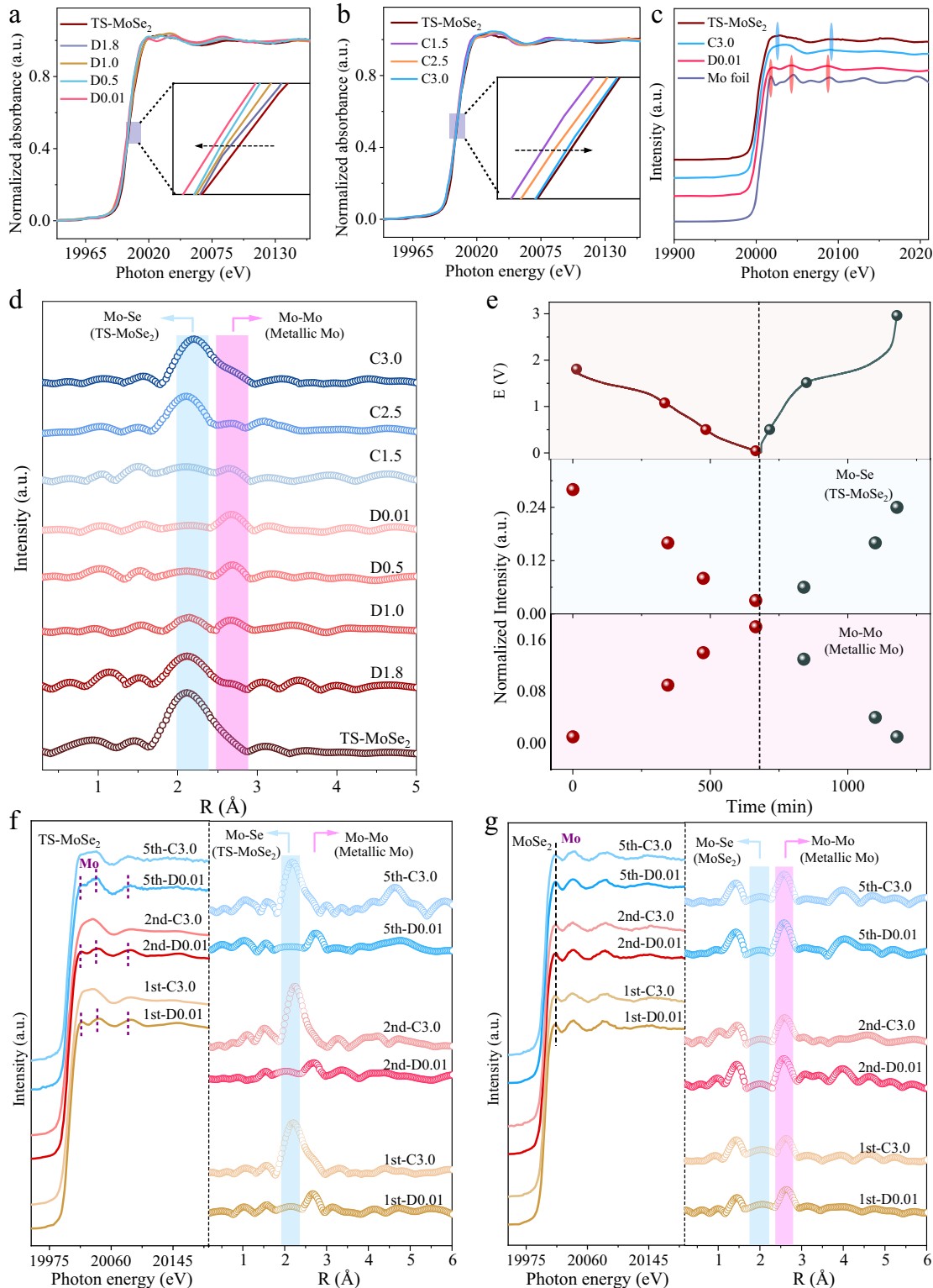

**Fig. 4 | Study on discharging and charging processes based on ex situ XAS.** Ex situ Mo K-edge XANES spectra of TS-MoSe$_2$ during the first **a** discharged and **b** charged states. **c** Mo K-edge XANES spectra of pristine TS-MoSe$_2$, fully discharged TS-MoSe$_2$ (D0.01), fully charged TS-MoSe$_2$ (C3.0), and Mo foil. **d** Evolution of Mo K-edge EXAFS during electrochemical cycling. **e** The intensity evolution of the Mo-Se peak in TS-MoSe$_2$ (2.11 Å, representing the concentration of TS-MoSe$_2$) and the Mo-Mo peak in metallic Mo (2.69 Å, representing the concentration of Mo) during electrochemical cycling. Mo K-edge XANES and EXAFS spectra of TS-MoSe$_2$ **f** and MoSe$_2$ **g** after the first, second, and fifth cycles.

0.09 mA h cm$^{-2}$), and that its Coulombic efficiencies are near 100% over 100 cycles, further implying the good cycling stability. TS-MoSe$_2$ also exhibits excellent rate performance. As shown in Fig. 5d, TS-MoSe$_2$ delivers reversible discharge capacities of 652, 604, 562, 533, 502, 460, and 408 mA h g$^{-1}$ at current densities of 0.05, 0.1, 0.2, 0.5, 1.0, 2.0, and 5.0 A g$^{-1}$, respectively. Notably, when the current rate is switched back to 0.05 A g$^{-1}$, the specific capacity recovers to about 665 mA h g$^{-1}$. The cycle and rate performances are superior to those of most of the

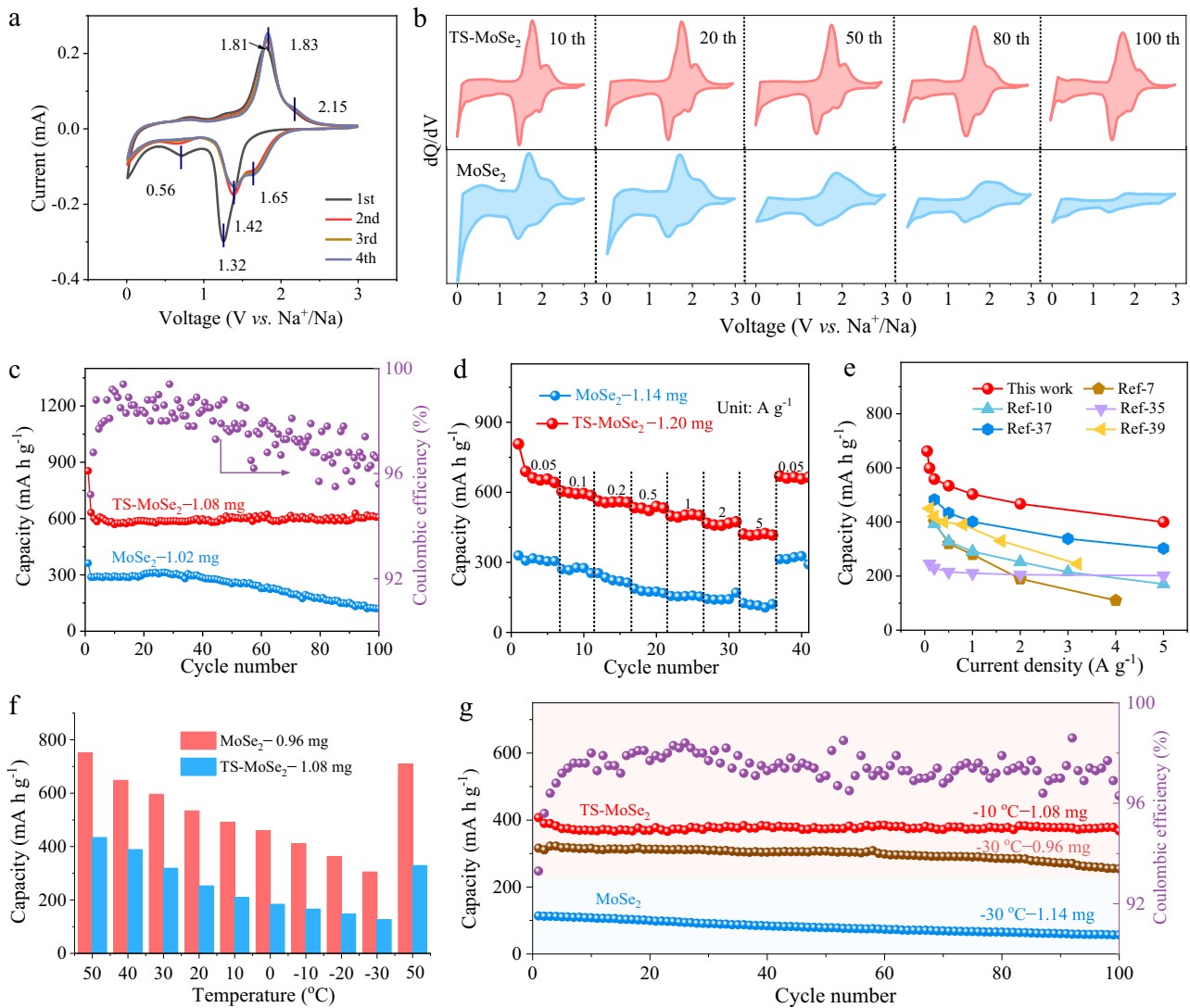

**Fig. 5 | Electrochemical performance. a** CV curves of TS-MoSe$_2$ between 0.01 and 3.0 V at a potential sweep speed of 0.1 mV s$^{-1}$. **b** dQ/dV plots of TS-MoSe$_2$ and MoSe$_2$. **c** Cycling and **d** rate performances of TS-MoSe$_2$ and MoSe$_2$. **e** Comparison of the rate capacities of TS-MoSe$_2$ with a series of reported MoSe$_2$-based anodes. **f** Cycling performances of TS-MoSe$_2$ and MoSe$_2$ at different temperatures from 50 to −30 °C. **g** Cycling performances of TS-MoSe$_2$ at −10, −30 °C and MoSe$_2$ at −30 °C.

reported MoSe$_2$-based nanomaterials (Fig. 5e). Additionally, we also tested the cycling stability and rate performance of the TS-MoSe$_2$ electrode with increased loadings of the active materials. As shown in Supplementary Fig. 33, only a slight capacity reduction is observed. Besides, even with relatively high mass ratios of the active materials (the mass ratios of the active materials: carbon: binder are 7:2:1 and 8:1:1, respectively), TS-MoSe$_2$ still exhibits good cycling and rate performance. Meanwhile, Fig. 5f exhibits the rate performance of TS-MoSe$_2$ in the temperature range of 50 to −30 °C was also tested. Surprisingly, TS-MoSe$_2$ displays admirable adaptability to the temperature. When the temperature is as low as −30 °C, TS-MoSe$_2$ still remains a reversible capacity as high as 380 mA h g$^{-1}$ at 0.1 A g$^{-1}$ after 100 cycles (Fig. 5g). In sharp contrast, the reversible capacity of MoSe$_2$ at −30 °C is only 128 mA h g$^{-1}$ after 100 cycles. In view of the superior sodium storage performance of TS-MoSe$_2$ in the half cells, the Na-ion full cells were further assembled with homemade Na$_3$V$_2$(PO$_4$)$_2$O$_2$F (NVPOF) as a cathode to preliminarily assess its practicability as an anode for SIBs (Supplementary Figs. 34, 35). As shown in Supplementary Fig. 35b, c, the full cells exhibit good cycling performance and the capacity can still maintain 434.9 mA h g$^{-1}$ after 200 cycles at 0.2 A g$^{-1}$ (based on the mass of the anode). The full cells also present superior rate capabilities (Supplementary Fig. 35d), in which about 70.2% of the capacity can be

retained even when the current density increases by 50-folds from 0.1 to 5 A g$^{-1}$. The good rate capabilities endow the full cells with a specific energy of 108.6 Wh kg$^{-1}$ at a power density of 19.0 W kg$^{-1}$, and even 74.1 Wh kg$^{-1}$ at a power density of 648.5 W kg$^{-1}$ (based on the total mass of the electrode materials), which are comparable or superior to those of many reported full cells (Supplementary Fig. 35e).

## Electrochemical kinetics analysis

To deeply understand the excellent reaction kinetics of TS-MoSe$_2$ as the anode for SIBs, the temperature-dependent electrochemical impedance spectroscopy (EIS) spectra of TS-MoSe$_2$ and MoSe$_2$ were investigated (Fig. 6a and Supplementary Fig. 36). The Nyquist plots exhibit a high-frequency semicircle and a low-frequency sloping line, which refer to the charge transfer resistance (R$_{ct}$) at the electrolyte interface and the Na$^+$ diffusion resistance in the electrode, respectively[53]. The fitted parameters of TS-MoSe$_2$ and MoSe$_2$ are shown in Supplementary Table 4. Obviously, the R$_{ct}$ values of TS-MoSe$_2$ are always lower than those of its counterparts under all the test temperatures, indicating that the tensile strain contributes to accelerating the electron transfer rate of TS-MoSe$_2$. Then, we further analyzed the diffusion in the low-frequency region by calculating the diffusion coefficient of Na$^+$ (D$_{Na}$) (the details can be found in

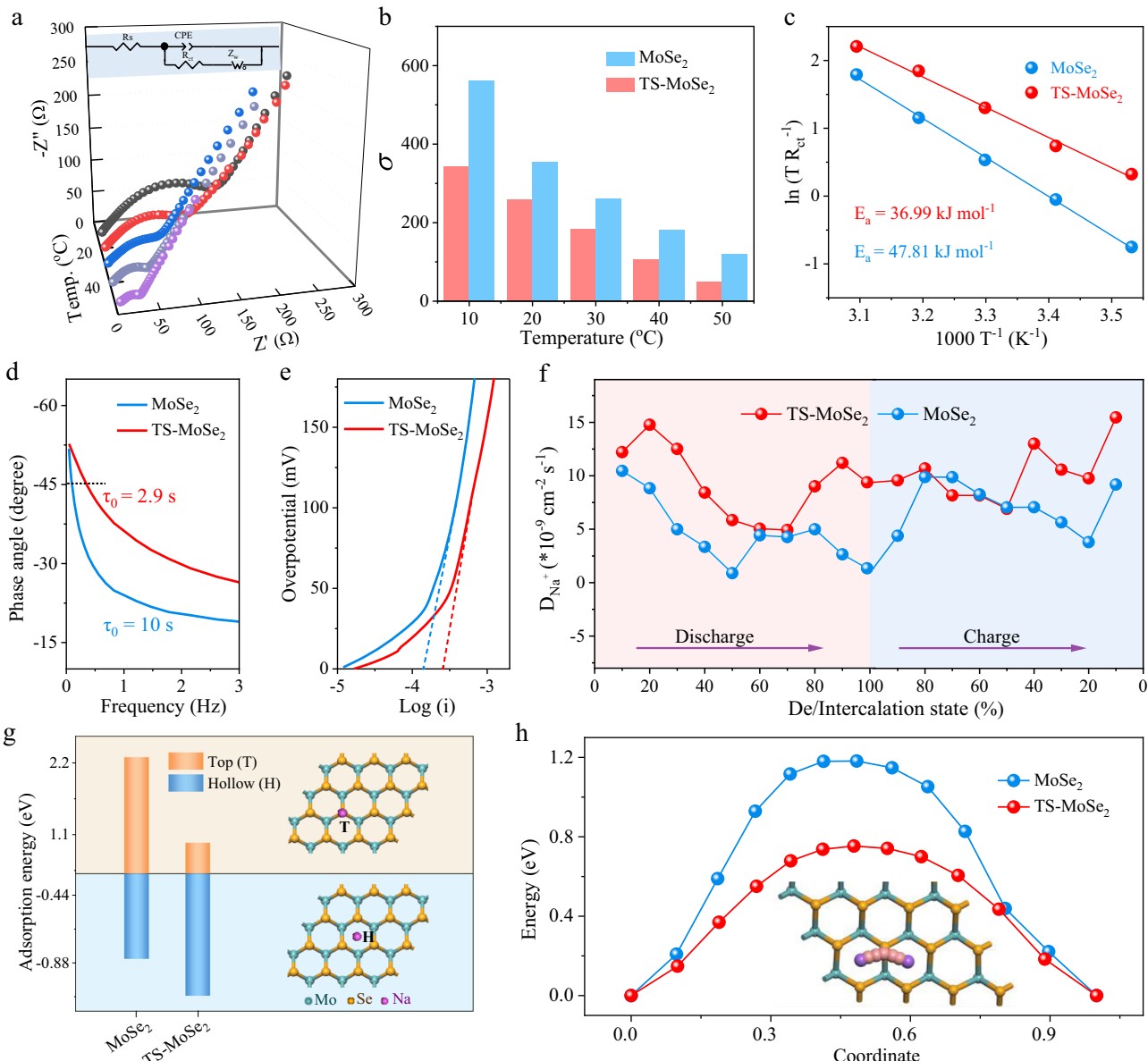

**Fig. 6 | Electrochemical kinetics analysis. a** EIS spectra of TS-MoSe$_2$ at different temperatures after 5 cycles. The inset is an equivalent circuit used to simulate EIS spectra. **b** σ values at different temperatures calculated from EIS curves. **c** Arrhenius plots of ln (T/R$_{ct}$) vs. 1/T in TS-MoSe$_2$ and MoSe$_2$ electrodes. **d** Bode plots of TS-MoSe$_2$ and MoSe$_2$ electrodes. **e** Tafel plots of TS-MoSe$_2$ and MoSe$_2$ electrodes during the anodic scan. **f** The sodium ion diffusion coefficient vs. De/Intercalation state of TS-MoSe$_2$ and MoSe$_2$ electrodes during the discharging/charging process after 5 cycles. **g** The adsorption sites of sodium ions and the corresponding adsorption energies. **h** The diffusion pathway of sodium ions and the corresponding diffusion energy barrier.

Supporting Information). The $D_{Na}$ is inversely proportional to the Warburg factor σ value and the σ can be obtained by fitting the real part Z′ of the electrochemical impedance spectroscopy with $\omega^{-1/2}$. As shown in Fig. 6b and Supplementary Fig. 37, TS-MoSe$_2$ exhibits a much smaller σ value than MoSe$_2$, suggesting its faster Na$^+$ diffusion rate. Moreover, based on the excellent ion diffusion kinetics of TS-MoSe$_2$, we further calculated its apparent activation energy ($E_a$) of sodium ion diffusion according to Arrhenius equations[54,55]. As displayed in Fig. 6c, the $E_a$ value of TS-MoSe$_2$ is determined to be 36.99 kJ mol$^{-1}$, which is smaller than that of MoSe$_2$, manifesting that the tensile strain could evidently lower the reaction activation energy and thus accelerate the reaction kinetics[56]. Besides, according to the correlation of the phase angle with the characteristic frequency, the corresponding time constant of the sample was also studied using the formula $\tau_0 = 1/f_0$, where $\tau_0$ is the minimum time required to release all the energy with an efficiency >50%. The smaller the value of $\tau_0$, the more conducive to

rapid ion diffusion and transmission, and $f_0$ is the characteristic frequency when the phase angle is −45°. As shown in Fig. 6d, the time constant of TS-MoSe$_2$ was calculated to be 2.9 s, which is significantly lower than that of MoSe$_2$ (10 s). The fast frequency response of TS-MoSe$_2$ further provides evidence for its smaller charge transfer resistance and better Na$^+$ diffusion/transportation dynamics[57,58].

Furthermore, Na$^+$ diffusion kinetics was also evaluated by the linear relationship between the redox peak current ($I_p$) and the sweep speed ($v^{1/2}$) based on the CV curves at different scan rates (Supplementary Fig. 38). According to the Randles−Sevcik formula, the slope of the fitted $I_p$ - $v^{1/2}$ is proportional to the $D_{Na}$ (see details in Supporting Information). As presented in Supplementary Fig. 39, the slopes of TS-MoSe$_2$ at the oxidation peak and reduction peak are 1.81 and −1.30, which are greater than those of MoSe$_2$ (1.24/−0.92), in accordance with those obtained from EIS. Furthermore, Tafel plots of TS-MoSe$_2$ and MoSe$_2$ were used to further study their reaction kinetics (Fig. 6e). As

the overpotential ($\eta$) approaches to zero, the plot deviates sharply from a linear behavior and can be extrapolated to an interception of log $i_0$. Based on the Butler Volmer model, the standard rate constant ($k_0$) of an electrochemical reaction is proportional to its exchange current ($i_0$)[59]. Clearly, TS-MoSe$_2$ displays a higher $i_0$ value during the anodic scan compared with MoSe$_2$, implying the faster oxidative kinetics of the TS-MoSe$_2$. The galvanostatic intermittent titration technique (GITT) was further performed to access the Na$^+$ diffusion kinetics of TS-MoSe$_2$ and MoSe$_2$ upon cycling (Fig. 6f and Supplementary Fig. 40)[60]. Clearly, the calculated $D_{Na}$ values of TS-MoSe$_2$ are larger than those of MoSe$_2$ at most of the discharging/charging states, while in some regions, their $D_{Na}$ values almost overlap. Based on the foregoing analyses, TS-MoSe$_2$ experiences the in-/de-tercalation and conversion reactions (generally, the former has higher $D_{Na}$ due to weaker interlayer van der Waals forces[3,61]), while for MoSe$_2$, Se/Na$_2$Se becomes the sole redox couple after the initial cycling that only occurs the conversion reaction (Se + 2Na$^+$ + 2e$^-$ $\leftrightarrow$ Na$_2$Se). Thus, the conversion process of the two cases involves similar intermediate phases, thereby resulting in almost the same $D_{Na}$ values.

To further confirm the influence of strain engineering on the diffusion kinetics of sodium ions, DFT calculations were carried out. Two typical adsorption sites were considered in Fig. 6g: the top of the Mo/Se atom, that is, the Top site (T); the hollow position in the center of the six-membered ring, which is the Hollow position (H). Then, the adsorption energy of Na$^+$ ($E_{ad}$) on TS-MoSe$_2$ and MoSe$_2$ was simulated in Fig. 6g, in which the positive $E_{ad}$ values at the T site suggest that the optimized adsorption site lies in the H site. Additionally, when adsorbed at the H position, the larger $E_{ad}$ of TS-MoSe$_2$ for sodium ions than MoSe$_2$ indicates that the tensile strain can increase the adsorption of the material to sodium ions. Finally, the diffusion path of sodium ions between two adjacent adsorption sites was simulated (the inset in Fig. 6h and Supplementary Fig. 41), and the corresponding diffusion energy barrier was calculated. From Fig. 6h, it can be seen that the diffusion energy barrier of sodium ions in TS-MoSe$_2$ is 0.75 eV, lower than that in MoSe$_2$. The calculation results manifest that the tensile strain can accelerate the dynamics of sodium ions by improving the adsorption energy to sodium ions as well as reducing its diffusion energy barrier, which is consistent with the above experimental results.

## Discussion

In summary, we have demonstrated that the constructed TS-MoSe$_2$ can transfer the strain gene to its discharged product Mo and uncovered the decisive role of the tensile strain in regulating Gibbs free energy change of the redox chemistry in the charging process to promote the efficient reversible conversion reaction. By electrochemical in situ Raman, ex situ XPS, and XAS, as well as DFT calculations, it was proved that the tensile strain could improve the activity of Mo, thus resulting in a reversible sodium storage mechanism, which endows TS-MoSe$_2$ with favorable reaction kinetics and thereby highly reversible capacity even in a wide temperature range. Our work provides insights into the electrochemical storage mechanism of conversion-type TMDs, which is essential in improving their electrochemical performance.

## Methods
### Materials
Ethanol ($\geq$99.7%) and methanol ($\geq$99. 5%) were obtained from Beijing Tongguang Fine Chemical Company. Phosphomolybdic acid hydrate (POM), 2-methylimidazole (2-MI), selenium powder, and bulk MoSe$_2$ were purchased from Aladdin reagents. Hydrazine hydrate ($\geq$80 wt%) was bought from Sinopharm Chemical Reagent Co., Ltd. All chemicals used in the experiments were analytical grade without further purification.

### Synthesis of Mo-precursor
In a typical synthesis, 0.03 mmol of phosphomolybdic acid hydrate (POM) was dissolved in 200 mL of ethanol. Then, 200 mL of ethanol containing 6.4 mmol of 2-methylimidazole (2-MI) was dropwise added into the above solution to form a yellow and transparent solution under vigorous stirring. After stirring for 24 h at room temperature, a light green precipitate was collected by centrifuging and washing with ethanol and then redispersed in 50 mL of ethanol/methanol (V/V = 4:1).

### Synthesis of TS-MoSe$_2$
Typically, 0.5 mmol of selenium powder was added to 2.0 mL of hydrazine hydrate (80 wt%) in a separate flask, then the colorless solution rapidly changed dark brown and was kept under atmospheric conditions at least for one night. Subsequently, the resultant hydrazine hydrate–Se solution was slowly added to the aforementioned Mo-precursor solution under vigorous stirring. After that, the mixed solution was transferred into a Teflon-lined stainless steel autoclave and kept at 180 °C for 12 h. After being cooled to room temperature naturally, the generated precipitate was centrifuged and washed with ethanol and water several times and then dried at 60 °C.

### Materials characterizations
The morphology and structure of the as-prepared products were characterized by field-emission scanning electron microscopy (FE-SEM; JSM-6490LV) and transmission electron microscopy (TEM; JEOL JEM-2010) as well as high-resolution transmission electron microscopy (HR-TEM). Elemental mapping images were recorded using energy-dispersive X-ray spectroscopy (EDX) attached to TEM. The crystal phase of the products was detected by powder X-ray diffraction (XRD, Bruker D8 Advance) in the range of 5−80° (2θ) with a scanning step of 10° min$^{-1}$. Raman spectrum was obtained on a Renishaw inVia Raman spectrometer with a laser of 532 nm. X-ray photoelectron spectroscopy (XPS) was measured on an ESCALAB 250 spectrometer (Perkin−Elmer). The elemental contents of C, H, and N were obtained on a Thermo Scientific Flash 2000 CHN-analyzer. Fourier-transform infrared (FT-IR) spectra were collected on a Bruker VECTOR 22 spectrometer. X-ray adsorption spectroscopy (XAS) measurements of the power samples were measured in transmission mode at the 1W1B station in Beijing Synchrotron Radiation Facility (BSRF).

Ex situ XPS characterizations: firstly, the battery was discharged and charged up to the required potential using a LAND workstation at a current density of 0.02 A g$^{-1}$. Then, the battery was disassembled in a glove box to collect the electrode sheet. Afterward, the resulting electrode sheet was washed with dimethyl carbonate (DMC) to remove any residual salts. Finally, the tested electrode sheet was transported with a vacuum transfer module from the glove box to the XPS test system to avoid component changes when exposed to air. Furthermore, the electrode sheet was also etched with an Ar$^+$ ion beam before the test to further avoid interference from surface SEI.

In situ Raman spectroscopy: The in situ Raman was measured using a Renishaw inVia Raman spectrometer with a laser of 532 nm. The electrochemical cells were adapted from CR2025 coin cells: a 4 mm hole was drilled in the top and then sealed with a thin cover glass slide with epoxy. The homemade coin cells were cycled at 100 mA g$^{-1}$ between 0.01 and 3.0 V.

Ex situ XAS: Electrodes at different discharge/charge conditions were disassembled and sealed in the glove box. All samples above were measured in fluorescence mode at ambient temperature. And the unreacted MoSe$_2$ detected in EXAFS spectra, which is inevitable, has been subtracted.

### Electrochemical measurements
The electrochemical behavior of the as-synthesized products was carried out by using two electrode CR2025 coin cells. All cells were assembled in an argon-filled glove box. The working electrode was

composed of 60 wt% active material, 30 wt% carbon black, and 10 wt% sodium carboxymethyl cellulose binder (CMC), which were mixed homogeneously with deionized water and the resultant slurry was pasted onto a copper foil current collector. Then the coated electrode was dried at 120 °C overnight. The thickness, diameter, and area of the electrode were 20–30 μm, 14 mm, and 1.54 cm$^{-2}$, respectively. The cells were assembled in an Ar-filled glove box and the mass loading of the active material is around 0.8–1.2 mg cm$^{-2}$. Sodium metal was used as counter and reference electrodes, and glass fiber paper (GF/C, Whatman, diameter: 16 mm) was used as the separator. The electrolyte was a solution of 1 M NaClO$_4$ in a 1:1 volume mixture of ethylene carbonate (EC)/dimethyl carbonate (DMC) to which 5 wt% fluoroethylene carbonate (FEC) was added. Galvanostatic charging-discharging curves of the cell were performed on a LAND CT2001A cell at different current densities in the voltage range of 0.01–3.00 V vs. Na$^+$/Na at room temperature. Furthermore, Electrochemical impedance spectroscopy (EIS) was performed on an electrochemical station (CHI-660) in the frequency range of 100 kHz to 0.01 Hz at the open circuit voltage. In order to effectively avoid errors introduced during the testing process, 3 specimens were tested for each type of battery performance evaluation to ensure that they have almost identical results. The electrochemical stability of the 2-MI molecule was evaluated in a three-electrode configuration using a glassy carbon electrode as the working electrode, a platinum sheet as the counter electrode, and Ag/AgCl (in saturated KCl aqueous solution) as the reference electrode. The potential vs. Ag/AgCl was converted into potentials vs. standard Na$^+$/Na, under the assumption that the potential of the Ag/AgCl electrode was 3.326 V vs. Na$^+$/Na. For the Na-ion full cells, the cathode was made of homemade NVPOF, carbon black, and poly(vinylidene fluoride) (PVDF) in a weight ratio of 8:1:1 on an aluminum foil. The mass loadings of TS-MoSe$_2$ and NVPOF were about 0.85 and 4.05 mg cm$^{-2}$, respectively. The capacity ratio of anode/cathode was controlled to be around 1.05:1. Meanwhile, the anode was electrochemically activated for three cycles before it was used in the full cells.

### Calculation details

Density functional theory (DFT) calculations were carried out with the Cambridge Sequential Total Energy Package (CASTEP)[62] based on the plane-wave-pseudopotential approach. The exchange and correlation interactions were described by the Perdew–Burke–Ernzerhof (PBE) functional combined with the generalized gradient approximation (GGA)[63]. The Grimme's semi-empirical DFT-D[64] correction was contained for the computations to ensure a better description of the electron interaction in a long range. The cut-off energy was set to 750 eV by using the Norm-conserving pseudopotential[65–67]. All atomic positions are fully relaxed during optimization (convergence thresholds of maximum displacement <0.001 Å, maximum force <0.03 eV/Å, and the energy difference <1.0 × 10$^{-5}$ eV/atom).

The linear response density functional perturbation theory (DFPT)[68–70] implemented in the CASTEP code was used to obtain the entropies and zero-point energy in the quasi-harmonic approximation. The Gibbs free energies for all the reactions were calculated at 298.15 K, and the calculation formula was defined as:

$$G = E_{DFT} - TS + E_{ZPE} \tag{6}$$

where $E_{DFT}$, TS, and $E_{ZPE}$ stand for the DFT energy, entropy contribution, and zero-point energy, respectively.

The diffusion of Na was evaluated by searching the plausible migration paths and identifying their transition states with the lowest diffusion energy barrier. Transition state searching calculations was performed according to the generalized synchronous transit method implemented in the CASTEP, in which the LST/QST algorithm combined the linear (LST) and quadratic synchronous transit (QST) methods with conjugate gradient (CG) refinements[71].

The adsorption energies ($E_{ad}$) for the Na atom on the TS-MoSe$_2$/MoSe$_2$ are calculated by the following equation:

$$E_{ad} = E_{MoSe_2/cluster-adsorbate} - E_{MoSe_2/cluster} - E_{adsorbate} \tag{7}$$

where $E$MoSe$_2$/Cluster-adsorbate stands for the total energy of Na adsorbed on the TS-MoSe$_2$/MoSe$_2$ and Na$_2$Se adsorbed on the Mo$_{15}$/Mo$_{15}$-MI clusters, $E$MoSe$_2$/Cluster represents the energy of the TS-MoSe$_2$/MoSe$_2$ and Mo$_{15}$/Mo$_{15}$-MI clusters, and $E$adsorbate is the energy of Na/Na$_2$Se.

## Data availability

All data generated in this study are provided in the Supplementary Information/Source Data file. Source data are provided with this paper.

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

## Acknowledgements

This work was financially supported by the National Natural Science Foundation of China (No. 21872008 and No. 22101023), the Natural Science Foundation of Beijing, China (Grant No. 2212019), and China Postdoctoral Science Foundation (No. 2021M690016). The authors would like to thank the Beijing Synchrotron Radiation Facility and Analysis & Testing Center of the Beijing Institute of Technology measurements.

## Author contributions

M.C., B.M., and M.J. conceived the idea, designed the experiments, analyzed the data, and wrote the paper. Y.H. performed computational modeling studies. Y.W. contributed to the in situ Raman test. Z.Y. conducted part of the electrochemical tests, electron microscopic experiments, and data analysis. T.M. participated in the materials preparation and data analysis. X.W. contributed to the electrochemical tests and data analysis. M.J. and Y.H. contributed equally to this work.

## Competing interests

The authors declare no competing interests.
