## [Peer Review File · Nature Communications]

Strain-regulated Gibbs free energy enables reversible redox chemistry of chalcogenides for sodium ion batteriesREVIEWER COMMENTS

Reviewer #1 (Remarks to the Author):

I had read through the manuscript "Strain-gene-regulated Gibbs Free Energy Enabling Reversible Redox Chemistry of Chalcogenides for Sodium Ion Batteries". This is an impressive work that shows the effect of strain-engineering on high reversible sodium-ion storage material. This enhancement on the electrochemical performance is remarkable. The manuscript is well organized with sufficient data. Therefore, I recommend a minor revision before the publication of this manuscript. Some specific comments are listed below:

1. The material information (e.g. synthesis process) and the related characterizations (e.g. SEM, TEM) are suggested.
2. The content of the 2-MI in the TS-MoSe₂ is suggested to be quantified. Such as identifying the N content and then calculating the content of 2-MI.
3. The BET surface area of MoSe₂ and TS-MoSe₂ is suggested to be given. It is interesting that the specific capacity is largely enhanced. What is the inside reason for the enhancement?
4. Will the presence of 2-MI affect the formation of SEI layers, which is significant in the reactions.
5. It is suggested that the authors place the right order of the figures, e.g in Figure 2 and 3.
6. Some related references are suggested, such as Nano-Micro Letters, 2021, 13, 55, and Energy Environ. Mater., 2020, 3: 221-234

Reviewer #2 (Remarks to the Author):

This work reports the effort of adding organic molecules to make the Na/MoSe reaction reversible. The mechanism was not clearly described and it should not be published in its current form unless the following issues are satisfactorily addressed:

1. Most importantly, we expect from the theoretical part a comparison of the reversible reaction with the other unwanted irreversible reaction. Should the reversible reaction be preferred over the irreversible one where Mo stays inert?
2. The experimental proof of the reversibility is not convincing enough. Was elemental Se observed? Was metallic Na observed? We see a study of Mo-Mo and Mo-Se bonds only. Should also supply the study of Na-Se, Se-Se, Na-Na and Na-Mo bonds so that the readers can have a clearer overall picture.
3. Some technical details should be fixed: The energy in Fig 1. was per atom? per formula unit? Regulate the unit. Details of Δ_G computation should be given. Was quasi-harmonic approximation used again? What temperature was assumed? Room temperature?
4. In addition, the mass loading of the active material is around 0.8–1.2 g cm⁻². The author should check it carefully. It should be mg cm⁻²? The loading is very low which will contribute to cycle stability and high rate. 30 wt% carbon black will help retard the side reaction.
- 5.1. The position of the absorption edge could indicate the valence state, and the oscillations in the XANES region could indicate the local structure too. It seems that the oscillations of C3.0 and TS-MoSe₂ are quite different in Fig. 4b, So, how to conclude that "Mo participates in the entire electrochemical process and undergoes a reversible sodium storage reaction". Furthermore, it is not convincing that the presence of metallic Mo could be determined by one peak in Fig. 4c. I think C3.0 in Fig. 4C also contains metallic Mo, as it has an obvious peak shown below.
6. The signal-to-noise ratio of XAFS seems very poor since there are many glitches at the white line peak of C3.0 in Fig. 4c. With such a poor signal-to-noise ratio, is the R space data credible? For example, the Mo-Mo peak is very low in D0.01, but the Mo-Mo peak in metallic Mo is very high. The highest peaks in D1.0 locate at $\sim 1.2\text{\AA}$ and 5\AA , which is due to the poor quality of the data in my opinion. So, the k-space data should be provided. Finally, supposing every claim was well supported, I would not consider this material's potential for actual applications in sodium-ion batteries because it still suffers from the stress and strain issues common to all conversion-type electrodes. Highly desirable research in this field should address this issue first.

Reviewer #3 (Remarks to the Author):

There are several published works on the topic of transition metal dichalcogenides for rechargeable alkali metal ion batteries. These materials tend to suffer from high first cycle loss, voltage hysteresis, considerable volume changes leading to capacity loss etc. In addition, high cost of such materials is a major barrier for use in low-cost sodium ion batteries. In the present work authors describe synthesis of a MoSe₂ type TMD, which under localized tensile deformation shows improved electrochemical performance as battery electrode over neat/pristine MoSe₂. The work seem to be interesting but has some major flaws that need to be rectified before publication in any scientific journal.

1. Fig 2f: what is the additional peak observed in TS-MoSe₂ close to 325 cm⁻¹? same for Figures g and j.

2. Fig 3b: The in-situ Raman is inconclusive. It is not uncommon to observe such subtle changes with-in the same sample from one location to another. Several different samples need to be tested to confirm the conclusions related to shift/suppression/re emergence of E_{2g1} band.

3. Fig.4: Caption reads sodium storage mechanisms--please explain what new mechanisms or electrochemical reactions are observed TS-MoSe₂?

4. Fig 5. Shows improved gravimetric capacity and rate capability of TS-MoSe₂ over MoSe₂. What is the mass of each electrode? Please supply areal capacity values. Also, mention how many specimens were tested for each type? In addition, without the 1st and 2nd cycle voltage profiles it is not possible to make any conclusions about superiority of TS MoSe₂ over MoSe₂.

5. There is barely any difference in the Na⁺ diffusion coefficient in TS-MoSe₂ over MoSe₂ in Fig 6f. The unit has a factor of 10⁻⁹. In certain regions [voltage values], the calculated values of the diffusion coefficient overlap.

6. Very sadly, the authors failed to cite the most important initial works on chalcogenides for Na⁺ batteries [first report on TMD for Na⁺ battery by Singh: <https://pubs.acs.org/doi/abs/10.1021/nn406156b>] and sustainable Na⁺ batteries [Nodal review article by Archer/Tarascon on Na⁺ batteries in Nature Chemistry: <https://www.nature.com/articles/nchem.2085>].

Overall, this is an interesting work with a fancy title [please consider removing terms like "strained-gene" from the title] that requires some additional experimental validation to support the conclusions about the superiority of TS-MoSe₂ over MoSe₂. The work is obviously of some fundamental nature but of little use from practicality of TMD-based materials in batteries.

Response to Reviewers' comments:

Reviewer #1:

I had read through the manuscript “Strain-gene-regulated Gibbs Free Energy Enabling Reversible Redox Chemistry of Chalcogenides for Sodium Ion Batteries&”. This is an impressive work that shows the effect of strain-engineering on high reversible sodium-ion storage material. This enhancement on the electrochemical performance is remarkable. The manuscript is well organized with sufficient data. Therefore, I recommend a minor revision before the publication of this manuscript. Some specific comments are listed below:

Thank you very much for your constructive comments. These professional comments and suggestions are very helpful for us to improve the quality of our manuscript. We have revised the manuscript in accordance with your suggestions. Thanks for your valuable suggestions once again. Our responses to your questions are below:

Question 1.

The material information (e.g. synthesis process) and the related characterizations (e.g. SEM, TEM) are suggested.

Response:

We greatly appreciate your valuable comments. The material information and the related characterizations have been provided in the revised manuscript. **(Please see page 21 in the revised manuscript and Supplementary Figure 5)**

Revised as follows:

Corresponding revisions on page 21 in the revised manuscript:

Materials: Ethanol ($\geq 99.7\%$) and methanol ($\geq 99.5\%$) were obtained from Beijing Tongguang Fine Chemical Company. Phosphomolybdic acid hydrate (POM), 2-methylimidazole (2-MI), selenium powder, and bulk MoSe_2 were purchased from Aladdin reagents. Hydrazine hydrate ($\geq 80 \text{ wt}\%$) was bought from Sinopharm Chemical Reagent Co., Ltd. All chemicals used in the experiments were analytical grade without further purification.

Supplementary Figure 5. FE-SEM images of the selenium powder (a), POM (b), 2-MI (c), and the synthesized Mo-precursor (d).

Question 2.

The content of the 2-MI in the TS-MoSe₂ is suggested to be quantified. Such as identifying the N content and then calculating the content of 2-MI.

Response:

We greatly appreciate your valuable comments. According to your professional suggestions, the content of the 2-MI in the TS-MoSe₂ has been calculated based on the N content originating from CHN elemental analysis in Supplementary Table 2. Specifically, based on the chemical formula of the 2-MI, C₄H₆N₂, the content of the 2-MI is calculated by the following equation,

$$\omega_{2\text{-MI}} \% = \frac{\omega_{\text{element N}} \times M_{2\text{-MI}}}{M_{\text{element N}} \times 2} \times 100$$

Where the $\omega_{2\text{-MI}}$ and $\omega_{\text{element N}}$ are the mass fraction of the 2-MI and the element nitrogen, respectively. $M_{2\text{-MI}}$ and $M_{\text{element N}}$ are the molar mass of the 2-MI (82 g/mol) and the element nitrogen (14 g/mol), respectively. And thus, the mass loading of 2-MI could be determined to be 4.10 wt.% ($\frac{1.40\%}{14 \times 2} \times 82 \times 100 = 4.10\%$). **(Please see page 8 in the revised manuscript and Supplementary Table 2)**

Supplementary Table 2. CHN analysis results of TS-MoSe₂.

Samples	Element		
	C (wt.%)	H (wt.%)	N (wt.%)
TS-MoSe ₂	2.22	1.51	1.40

Question 3.

The BET surface area of MoSe₂ and TS-MoSe₂ is suggested to be given. It is interesting that the specific capacity is largely enhanced. What is the inside reason for

the enhancement?

Response :

We are very grateful for your valuable comments. Based on your professional suggestions, we used Brunauer–Emmett–Teller method to analyze the specific surface area and it can be seen that TS-MoSe₂ was determined to be 17.68 m² g⁻¹, which is slightly higher than that of the unstrained MoSe₂ (14.17 m² g⁻¹) (Supplementary Fig. 8). This small difference between these two samples in BET surface area hardly leads to a nearly twofold increase in specific capacity. Thus, compared with MoSe₂, the reversible storage of Na⁺ and relatively higher Na⁺ storage kinetics in TS-MoSe₂ may be the major reason for its higher specific capacity. **(Please see page 6 in the revised manuscript and Supplementary Fig. 8)**

Supplementary Figure 8. The nitrogen adsorption-desorption isotherm curves of TS-MoSe₂ and MoSe₂.

Question 4.

Will the presence of 2-MI affect the formation of SEI layers, which is significant in the reactions.

Response :

Thanks for your good question. We conducted the linear sweep voltammetry (LSV) test through the three-electrode system to further confirm the stability of the 2-MI molecule. As shown in Supplementary Fig. 15, no visible reduction peak ascribed to the 2-MI molecule was found within the operating voltage window, indicating that the 2-MI is not involved in the formation of the SEI layer. **(Please see pages 9 and 23 in the revised manuscript and Supplementary Fig. 15)**

Revised as follows:

Corresponding revisions on page 9 in the revised manuscript:

Furthermore, the stability of the 2-MI molecule was further confirmed by linear sweep voltammetry (LSV) curves, in which no visible reduction peak ascribed to the 2-MI molecule was found within the operating voltage window **(Supplementary Fig. 15)**.

Supplementary Figure 15. LSV curves of the electrolyte with 2-MI and 2-MI-free.

Corresponding revisions on page 23 in the revised manuscript:

The electrochemical stability of the 2-MI molecule was evaluated in a three-electrode configuration using glassy carbon electrode as the working electrode, platinum sheet as the counter electrode, and Ag/AgCl (in saturated KCl aqueous solution) as the reference electrode. The potential *vs.* Ag/AgCl was converted into potential *vs.* standard Na⁺/Na, under the assumption that the potential of the Ag/AgCl electrode was 3.326 V *vs.* Na⁺/Na.

Question 5.

It is suggested that the authors place the right order of the figures, e.g in Figure 2 and 3.

Response:

Thank you for your suggestion, we have adjusted the layout of Fig. 2 and 3 in the revised manuscript. **(Please see pages 7 and 11 in the revised manuscript)**

Question 6.

Some related references are suggested, such as Nano-Micro Letters, 2021, 13, 55, and Energy Environ. Mater., 2020, 3: 221-234.

Response:

Thank you for your suggestion and the useful references. The related references have been added in the revised manuscript. **(Please see ref. 5 and 9 in the revised manuscript)**

Reviewer #2:

This work reports the effort of adding organic molecules to make the Na/MoSe₂ reaction reversible. The mechanism was not clearly described and it should not be published in its current form unless the following issues are satisfactorily addressed:

Thank you very much for your constructive comments. These professional comments and suggestions are very helpful for us to improve the quality of our manuscript. We have revised the manuscript in accordance with your suggestions. Thanks for your valuable suggestions once again. Our responses to your questions are below:

Question 1.

Most importantly, we expect from the theoretical part a comparison of the reversible reaction with the other unwanted irreversible reaction. Should the reversible reaction be preferred over the irreversible one where Mo stays inert?

Response:

We greatly appreciate your valuable comments. Following your suggestion, we added the calculation of the Gibbs free energy of the irreversible reaction ($\text{Mo} + \text{Na}_2\text{Se} \rightarrow \text{Mo} + \text{Se} + 2\text{Na}^+ + 2\text{e}^-$, $\Delta G = 3.76 \text{ eV}$) for the unstrained Mo in Supplementary Fig. 3, which is lower than that of the reversible one ($\Delta G = 3.95 \text{ eV}$), indicating that the irreversible reaction may occur preferentially than the reversible one for the unstrained Mo. (Please see page 5 in the revised manuscript and Supplementary Fig. 2,3)

Revised as follows:

Corresponding revisions on page 5 in the revised manuscript:

Furthermore, for the unstrained Mo, this ΔG value corresponding to the reversible reaction is higher than that of the irreversible reaction ($\text{Mo} + \text{Na}_2\text{Se} \rightarrow \text{Mo} + \text{Se} + 2\text{Na}^+ + 2\text{e}^-$) (Supplementary Fig. 3), indicating that the irreversible reaction may occur preferentially than the reversible one for the unstrained Mo.

Supplementary Figure 2. The optimized structural model of (c) Se for the calculations of ΔG .

Supplementary Figure 3. The ΔG values per formula unit of two different reaction pathways for the discharged products of unstrained MoSe₂ (Mo and Na₂Se) during the charging process.

Question 2.

The experimental proof of the reversibility is not convincing enough. Was elemental Se observed? Was metallic Na observed? We see a study of Mo-Mo and Mo-Se bonds only. Should also supply the study of Na-Se, Se-Se, Na-Na and Na-Mo bonds so that the readers can have a clearer overall picture.

Response:

We greatly appreciate your constructive comments. Indeed, for MoSe₂, in addition to the Mo-Mo and Mo-Se bonds, Se/Na-related bonds should also be discussed to gain insight into the sodium storage process of MoSe₂. However, unfortunately, we have overlooked this point in the original manuscript, for which we sincerely apologize.

Thank you again for your valuable comments. Based on your professional suggestions, we further analyzed the corresponding Se K-edge EXAFS of TS-MoSe₂ and the results also confirmed the gradual transformation of Se-Mo bond (MoSe₂) to Se-Na bond (Na₂Se) upon discharging (Supplementary Fig. 23,24e). In the subsequent charging process, the Se-Na peak gradually disappears, while Se-Mo and Se-Se (TS-MoSe₂) peaks become stronger. These observations also proved the excellent electrochemical reversibility of TS-MoSe₂. In addition, the Se K-edge EXAFS spectra of TS-MoSe₂ at D0.01 and C3.0 during the second and fifth cycles were also recorded, further confirming the reversible conversion of TS-MoSe₂ in the subsequent cycles. (Supplementary Fig. 26,27). Notably, neither elemental Se nor metallic Na was detected during the whole discharging and charging processes of TS-MoSe₂, which is consistent with its reversible sodium storage process ($\text{MoSe}_2 + \text{Na}^+ \leftrightarrow \text{Mo} + \text{Na}_2\text{Se}$). Besides, due to the special “sandwich structure” of MoSe₂, the intercalated Na⁺ is located in the middle of the two layers of Se and is far away from Mo. Therefore, the Na-Mo bond can hardly be detected. **(Please see page 14 in the revised manuscript and Supplementary Fig. 23,24e,26,27)**

Furthermore, *ex-situ* XPS was also conducted to probe the structural evolution during the electrochemical processes. Specifically, during the whole evolution processes of discharging and charging, ten voltages were selected to evaluate the structural transformation of the TS-MoSe₂ electrode. As shown in the Mo 3d XPS spectra (Fig. 3a), at the beginning of the discharging process (1.8 and 1.5 V), two main characteristic peaks at 228.83 and 231.93 eV that are related to 3d_{5/2} and 3d_{3/2} of Mo⁴⁺ in MoSe₂ slightly shift towards the low binding energy, indicating the formation of the Na_xMoSe₂ intermediate. With further discharging (1.0 and 0.4 V), a new component with lower binding energies at 227.43 (Mo 3d_{5/2}) and 230.53 eV (Mo 3d_{3/2}) appears and it can be

assigned to metallic Mo³⁸, suggesting that the Na_xMoSe₂ has partly transformed into the metallic Mo. At fully discharged state, the Na_xMoSe₂ completely disappears and only metallic Mo is detected. Correspondingly, the Se 3d peak at 54.5 eV first shifts to higher binding energy, and then restores to the original position, manifesting that Na₂Se finally forms through the polyselenide Na₂(Se)_{1+n} (n > 1) during the discharging process (Fig. 3c)³⁹. Afterwards, in the following charging process, the peaks of both Mo 3d and Se 3d core levels can be fully recovered to their pristine state for TS-MoSe₂, and in contrast, for unstrained MoSe₂, the metallic Mo is always present and meanwhile the elemental Se is eventually generated (Supplementary Fig. 17a,18a). These changes can be observed more visually in corresponding 2D mapping images of the Mo 3d and Se 3d XPS spectra (Fig. 3b and Supplementary Fig. 17b,18b,19), which demonstrate that the strain engineering enables TS-MoSe₂ to follow highly reversible sodium storage mechanism in the discharging and charging processes. **(Please see pages 10 and 22 in the revised manuscript, Fig. 3 and Supplementary Fig. 17,18,19)**

Revised as follows:

Corresponding revisions on page 14 in the revised manuscript:

Similarly, the corresponding Se K-edge EXAFS spectra (**Supplementary Fig. 24e**) also witnessed the gradual transformation of Se-Mo bond (MoSe₂) to Se-Na bond (Na₂Se) upon discharging. In the subsequent charging process, the Mo-Mo (metallic Mo) and Se-Na peaks gradually disappear, while Mo-Se/Se-Mo and Se-Se (TS-MoSe₂) peaks become stronger. These observations further prove the excellent electrochemical reversibility of TS-MoSe₂. In addition, the Mo and Se K-edge XANES and EXAFS spectra of TS-MoSe₂ at D0.01 and C3.0 during the second and fifth cycles were also recorded, further confirming the reversible conversion of TS-MoSe₂ in the subsequent cycles (**Fig. 4f and Supplementary Fig. 25-27**).

Supplementary Figure 23. The k-space XANES spectra of TS-MoSe₂ during the first discharge (b-e) and charge (f-h) state for Se K-edge.

Supplementary Figure 24. (e) *Ex-situ* Se K-edge EXAFS spectra of TS-MoSe₂ during the first discharging and charging states.

Supplementary Figure 26. The k-space XANES spectra of TS-MoSe₂ after the second (a,b) and fifth (c,d) cycles for Se K-edge.

Supplementary Figure 27. Se K-edge XANES and EXAFS spectra of TS-MoSe₂ after the first, second, and fifth cycles.

Corresponding revisions on page 10 in the revised manuscript:

Inspired by the positive influence that the strain engineering has achieved on the redox reaction by the DFT calculations, we first performed *ex-situ* XPS measurements to investigate the effect of the tensile strain on the sodium storage process. During the whole evolution processes of discharging and charging, ten voltages were selected to evaluate the structural transformation of the TS-MoSe₂ electrode. As shown in the Mo 3d XPS spectra (Fig. 3a), at the beginning of the discharging process (1.8 and 1.5 V), two main characteristic peaks at 228.83 and 231.93 eV that are related to 3d_{5/2} and 3d_{3/2} of Mo⁴⁺ in MoSe₂ slightly shift towards the low binding energy, indicating the

formation of the Na_xMoSe_2 intermediate. With further discharging (1.0 and 0.4 V), a new component with lower binding energies at 227.43 (Mo 3d_{5/2}) and 230.53 eV (Mo 3d_{3/2}) appears and it can be assigned to metallic Mo³⁸, suggesting that the Na_xMoSe_2 has partly transformed into the metallic Mo. At fully discharged state, the Na_xMoSe_2 completely disappears and only metallic Mo is detected. Correspondingly, the Se 3d peak at 54.5 eV first shifts to higher binding energy, and then restores to the original position, manifesting that Na_2Se finally forms through the polyselenide $\text{Na}_2(\text{Se})_{1+n}$ ($n > 1$) during the discharging process (Fig. 3c)³⁹. Afterwards, in the following charging process, the peaks of both Mo 3d and Se 3d core levels can be fully recovered to their pristine state for TS-MoSe₂, and in contrast, for unstrained MoSe₂, the metallic Mo is always present and meanwhile the elemental Se is eventually generated (Supplementary Fig. 17a,18a). These changes can be observed more visually in corresponding 2D mapping images of the Mo 3d and Se 3d XPS spectra (Fig. 3b and Supplementary Fig. 17b,18b,19), which demonstrate that the strain engineering enables TS-MoSe₂ to follow highly reversible sodium storage mechanism in the discharging and charging processes.

Fig. 3. Study on discharging and charging processes based on *ex-situ* XPS spectra. a-c *ex-situ* Mo 3d XPS spectra (a) and corresponding mapping image (b), as well as Se 3d XPS spectra (c) of TS-MoSe₂ during the initial discharging and charging processes.

Supplementary Figure 17. *Ex-situ* Mo 3d XPS spectra of MoSe₂ during the initial discharging and charging processes (a) as well as the corresponding mapping image (b).

Supplementary Figure 18. *Ex-situ* Se 3d XPS spectra (a) and corresponding mapping image (b) of MoSe₂ during the initial discharging and charging processes.

Supplementary Figure 19. *Ex-situ* Se 3d XPS mapping image of TS-MoSe₂ during the initial discharging and charging processes.

Supplementary references:

38. Hu X., Zhu R., Wang B., Wang H. & Liu X. Sn catalyst for efficient reversible conversion between MoSe₂ and Mo/Na₂Se for high-performance energy storage. *Chem. Eng. J.* **440**, 135819 (2022).
39. Karger Z., et al. Selenium and selenium-sulfur cathode materials for high-energy rechargeable magnesium batteries. *J. Power Sources* **323**, 213–219 (2016).

Corresponding revisions on page 22 in the revised manuscript:

Ex-situ XPS characterizations: firstly, the battery was discharged and charged up to the required potential using a LAND workstation at a current density of 0.02 A g^{-1} . Then, the battery was disassembled in a glove box to collect the electrode sheet. Afterwards, the resulting electrode sheet was washed with dimethyl carbonate (DMC) to remove any residual salts. Finally, the tested electrode sheet was transported with a vacuum-transfer-module from the glove box to the XPS test system to avoid component changes when exposed to air. Furthermore, the electrode sheet was also etched with Ar^+ ion beam before the test to further avoid interference from surface SEI.

Question 3.

Some technical details should be fixed: The energy in Fig 1. was per atom? per formula unit? Regulate the unit. Details of delta G computation should be given. Was quasi-harmonic approximation used again? What temperature was assumed? Room temperature?

Response :

Thanks for your valuable comments. We are sorry that we didn't provide a very clear explanation about the details of the theoretical calculations. The energy calculations in Fig. 1c,e,g were based on per 2-MI molecule, per formula unit, and per Na_2Se molecule, respectively, which have been marked in the corresponding captions. Additionally, the specific calculation details for ΔG based on the formula of $G = E_{\text{DFT}} - \text{TS} + E_{\text{ZPE}}$ have been provided in Supplementary Table 1. Furthermore, in the calculation process, quasi-harmonic approximation was used to obtain the entropies and zero-point energy, and the Gibbs free energies for all the reactions were calculated at 298.15 K, which have been added in the Calculation details. **(Please see page 24 in the revised manuscript, Fig. 1c,e,g, and Supplementary Table 1)**

Revised as follows:

Fig. 1. Theoretical calculations. **c** The adsorption energies per 2-MI molecule on MoSe₂ and Mo. **e** The ΔG values per formula unit of the reaction between $\text{Na}_2\text{Se}/\text{Mo}$ and MoSe₂ under strained and unstrained conditions. **g** The adsorption energies of per Na_2Se molecule on TS-Mo and Mo and the corresponding Na-Se distance evolution.

Supplementary Table 1. The detailed calculation data of free energies during the reaction.

Reaction	ΔG (eV)	ΔE (eV)	ZPE (eV)	T ΔS (eV)
TS-Mo + 2Na ₂ Se \rightarrow TS-MoSe ₂ + 4Na	3.0608	3.0757	0.0417	0.0566
Mo + Na ₂ Se \rightarrow MoSe ₂ + 4Na	3.9507	3.9302	0.0517	0.0312
Mo + Na ₂ Se \rightarrow Mo + Se + 2Na	3.7609	3.7929	-0.0016	0.0304

Corresponding revisions on page 24 in the revised manuscript:

The linear response density functional perturbation theory (DFPT)⁶⁹⁻⁷¹ implemented in the CASTEP code was used to obtain the entropies and zero-point energy in the quasi-harmonic approximation. The Gibbs free energies for all the reactions were calculated at 298.15 K, and the calculation formula was defined as $G = E_{\text{DFT}} - TS + E_{\text{ZPE}}$, where E_{DFT} , TS, and E_{ZPE} stand for the DFT energy, entropy contribution, and zero-point energy, respectively.

Supplementary references:

69. Baroni, S., Gironcoli, S. & Corso, A. Phonons and related crystal properties from density-functional perturbation theory. *Rev. Mod. Phys.* **73**, 515–562 (2001).
70. Gonze, X. & Lee, C. Dynamical matrices, born effective charges, dielectric permittivity tensors, and interatomic force constants from density-functional perturbation theory. *Phys. Rev. B* **55**, 10355–10368 (1997).
71. Refson, K., Tulip, P. & Clark, S. Variational density-functional perturbation theory for dielectrics and lattice dynamics. *Phys. Rev. B* **73**, 155114 (2006).

Question 4.

In addition, the mass loading of the active material is around 0.8-1.2 g cm⁻². The author should check it carefully. It should be mg cm⁻²? The loading is very low which will contribute to cycle stability and high rate. 30 wt% carbon black will help retard the side reaction.

Response:

Thanks for your valuable comments. First of all, we are very sorry for our carelessness. In this revision, we have corrected the unit of mass loading. Subsequently, in order to clarify the effects of mass loading and the content of carbon black on cycling and rate performance, we further tested cells with high loadings, low carbon content and simultaneously increasing loading and decreasing carbon content, respectively (Supplementary Fig. 33). Indeed, as you said, the cycle and rate performance of TS-MoSe₂ and MoSe₂ do show some degradation compared to before, which may be

related to the electrolyte wettability and conductivity in the thick electrode. Nevertheless, the overall performance of TS-MoSe₂ is still significantly superior to that of MoSe₂. (Please see page 17 in the revised manuscript and Supplementary 33)

Revised as follows:

Corresponding revisions on page 17 in the revised manuscript:

Additionally, we also tested the cycling stability and rate performance of TS-MoSe₂ electrode with increased loadings of the active materials. As shown in **Supplementary Fig. 33**, only slight capacity reduction is observed. Besides, even with relatively high mass ratios of the active materials (the mass ratios of the active materials: carbon: binder are 7:2:1 and 8:1:1, respectively), TS-MoSe₂ still exhibits good cycling and rate performance.

Supplementary Figure 33. Cycling and rate performances of TS-MoSe₂ and MoSe₂ with increased loadings of the active materials (a,b), different mass ratios of active materials, carbon black and binder (c,d), as well as both increased loading and decreased carbon black content (e,f), respectively.

Question 5.

The position of the absorption edge could indicate the valence state, and the oscillations in the XANES region could indicate the local structure too. It seems that the oscillations of C3.0 and TS-MoSe₂ are quite different in Fig. 4b, So, how to

conclude that “Mo participates in the entire electrochemical process and undergoes a reversible sodium storage reaction”. Furthermore, it is not convincing that the presence of metallic Mo could be determined by one peak in Fig. 4c. I think C3.0 in Fig. 4C also contains metallic Mo, as it has an obvious peak shown below.

Response:

Thanks for your professional comments. We are so sorry for the unreasonable conclusion that was given in our original manuscript only from the XANES wiggle/oscillatory features of the post-edge region. Indeed, as you said, it seems that the oscillations of C3.0 and TS-MoSe₂ are quite different in Fig. 4c, which may be due to the amorphous nature of the regenerated MoSe₂ after full charging (C3.0) and poor signal-to-noise ratio. Furthermore, it is also not convincing to conclude the presence of metallic Mo from only one peak in Fig. 4c of the original manuscript.

In order to provide more sufficient evidence to prove the regeneration of MoSe₂ after charging and the formation of metallic Mo during the discharge process, we further increased the mass loading of the active component of the tested electrode sheets to improve the signal-to-noise ratio according to the suggestion of the XAS testing engineer. And meanwhile, we also provided *ex-situ* XPS to further monitor the sodium storage process of TS-MoSe₂.

Specifically, on the one hand, we increased the mass loading of the active component (about 3.0 mg) of the tested electrode sheets to improve the signal-to-noise ratio (the mass ratio of active material: carbon black: sodium carboxymethyl cellulose (CMC) binder: styrene butadiene rubber (SBR) binder = 8:1:0.5:0.5, the combination of CMC and SBR could effectively prevent the active component peeling off from the electrode). And the XAS spectra at potentials of D0.01 and C3.0 were retested to replace the data with relatively poor signal-to-noise ratio in the original manuscript. Upon being fully discharged to 0.01 V (D0.01), it can be noticed that in addition to the peak at 20013.2 eV attributed to Mo foil that becomes more pronounced compared to the original spectrum, other Mo foil feature peaks (20040.3 and 20084.1 eV) were also observed. In contrast, after the full charging (C3.0), the aforesaid peaks almost recover to the original state of TS-MoSe₂, while the Mo foil related features disappear. It should be noted that the Mo K-edge XANES spectra of the formed metallic Mo and the regenerated MoSe₂ are slightly different from those of corresponding Mo foil and pristine TS-MoSe₂, which may be caused by the ligand effect of imidazole and amorphous nature, respectively^{41,42}. **(Please see page 12 in the revised manuscript**

and Fig. 4c)

On the other hand, we further provided *ex-situ* XPS to confirm the formation of metallic Mo after discharging and regeneration of MoSe₂ after charging. During the whole evolution processes of discharging and charging, ten voltages were selected to evaluate the structural transformation of the TS-MoSe₂ electrode. As shown in the Mo 3d XPS spectra (Fig. 3a), at the beginning of the discharging process (1.8 and 1.5 V), two main characteristic peaks at 228.83 and 231.93 eV that are related to 3d_{5/2} and 3d_{3/2} of Mo⁴⁺ in MoSe₂ slightly shift towards the low binding energy, indicating the formation of the Na_xMoSe₂ intermediate. With further discharging (1.0 and 0.4 V), a new component with lower binding energies at 227.43 (Mo 3d_{5/2}) and 230.53 eV (Mo 3d_{3/2}) appears and it can be assigned to metallic Mo³⁸, suggesting that the Na_xMoSe₂ has partly transformed into the metallic Mo. At fully discharged state, the Na_xMoSe₂ completely disappears and only metallic Mo is detected. Afterwards, in the following charging process, the peaks of Mo 3d core level can be fully recovered to their pristine state for TS-MoSe₂. These changes can be observed more visually in the corresponding 2D mapping images of the Mo 3d XPS spectra (Fig. 3b), which demonstrate that the strain engineering enables TS-MoSe₂ to follow highly reversible sodium storage mechanism in the charging and discharging processes. **(Please see page 10 in the revised manuscript and Fig. 3)**

Revised as follows:

Corresponding revisions on page 12 in the revised manuscript:

In addition, the corresponding wiggle/oscillatory features of the post-edge region of the pristine TS-MoSe₂, fully discharged TS-MoSe₂ (D0.01), fully charged TS-MoSe₂ (C3.0), and Mo foil can also reflect the variation in the local structure of TS-MoSe₂ during the electrochemical process. **(Fig. 4c)**. Upon being fully discharged to 0.01 V, the appearance of the fingerprint feature of Mo foil at 20013.2, 20040.3, and 20084.1 eV supports the formation of metallic Mo¹⁶. In contrast, after the full charging, the aforesaid peaks almost recover to the original state of TS-MoSe₂, while the Mo foil related features disappear, indicating that the discharging and charging processes of TS-MoSe₂ during the initial cycle are nearly fully reversible. It should be noted that the Mo K-edge XANES spectra of the formed metallic Mo and the regenerated MoSe₂ are slightly different from those of corresponding Mo foil and pristine TS-MoSe₂, which may be caused by the ligand effect of imidazole and amorphous nature, respectively

Fig. 4. Study on discharging and charging processes based on *ex-situ* XAS. c Mo K-edge XANES spectra of pristine TS-MoSe₂, fully discharged TS-MoSe₂ (D0.01), fully charged TS-MoSe₂ (C3.0), and Mo foil.

Supplementary references:

41. Swilem Y. & Al-Otaibi H. Structural studies of nucleation and growth of Cu and Fe nanoparticles using XAFS simulation. *AIMS Materials Science* **7**, 1–8 (2020).
42. Krbal M., *et al.* Amorphous-to-crystal transition in quasi-two-dimensional MoS₂: implications for 2D Electronic Devices. *ACS Appl. Nano Mater.* **4**, 8834–8844 (2021).

Corresponding revisions on page 10 in the revised manuscript:

As shown in the Mo 3d XPS spectra (Fig. 3a), at the beginning of the discharging process (1.8 and 1.5 V), two main characteristic peaks at 228.83 and 231.93 eV that are related to 3d_{5/2} and 3d_{3/2} of Mo⁴⁺ in MoSe₂ slightly shift towards the low binding energy, indicating the formation of the Na_xMoSe₂ intermediate. With further discharging (1.0 and 0.4 V), a new component with lower binding energies at 227.43 (Mo 3d_{5/2}) and 230.53 eV (Mo 3d_{3/2}) appears and it can be assigned to metallic Mo³⁸, suggesting that the Na_xMoSe₂ has partly transformed into the metallic Mo. At fully discharged state, the Na_xMoSe₂ completely disappears and only metallic Mo is detected. Afterwards, in the following charging process, the peaks of Mo 3d core level can be fully recovered to their pristine state for TS-MoSe₂. These changes can be observed more visually in corresponding 2D mapping images of the Mo 3d and XPS spectra (Fig. 3b), which demonstrate that the strain engineering enables TS-MoSe₂ to follow highly reversible sodium storage mechanism in the discharging and charging processes.

Fig. 3. Study on discharging and charging processes based on *ex-situ* XPS spectra. a-c *ex-situ* Mo 3d XPS spectra (a) and corresponding mapping image (b), as well as Se 3d XPS spectra (c) of TS-MoSe₂ during the initial discharging and charging processes.

Question 6.

The signal-to-noise ratio of XAFS seems very poor since there are many glitches at the white line peak of C3.0 in Fig. 4c. With such a poor signal-to-noise ratio, is the R space data credible? For example, the Mo-Mo peak is very low in D0.01, but the Mo-Mo peak in metallic Mo is very high. The highest peaks in D1.0 locate at $\sim 1.2\text{\AA}$ and 5\AA , which is due to the poor quality of the data in my opinion. So, the k-space data should be provided.

Response:

Thank you very much for your professional advice. Indeed, as you said, the signal-to-noise ratios at C3.0, D0.01, and D1.0 V (including C2.5 V for the first cycle, as well as the C3.0 and D0.01 V for the second cycle) are weaker than those at the other potentials. Then, by consulting with the engineer, we found that the poor signal-to-noise ratio may arise from the relatively low mass loading ($0.8\text{-}1.2\text{ mg cm}^{-2}$) and the differences in structure, composition, and content of the species transformed under electrochemical conditions. To improve the signal-to-noise ratio of XAS data of these samples, we tried to increase the mass loading of the active component (about 3.0 mg) of the tested electrode (the mass ratio of active material: carbon black: CMC binder: SBR binder = $8:1:0.5:0.5$, the combination of CMC and SBR could effectively prevent the active component peeling off from the electrode). And the XAS spectra at potentials of C3.0, D0.01, D1.0, and C2.5 for the first cycle, as well as C3.0 and D0.01 V for the second cycle were retested to replace the data with relatively poor signal-to-noise ratio

in original manuscript. As expected, spectra with relatively stronger signals were obtained (Supplementary Fig. 21,22,25). Specifically, as shown in Fig. 4c, upon being fully discharged to 0.01 V, the appearance of the fingerprint feature of Mo foil at 20013.2, 20040.3, and 20084.1 eV supports the formation of metallic Mo¹⁶. In contrast, after the full charging, the aforesaid peaks almost recover to the original state of TS-MoSe₂, while the Mo foil related features disappear, indicating that the discharging and charging processes of TS-MoSe₂ during the initial cycle are nearly fully reversible. It should be noted that the Mo K-edge XANES spectra of the formed metallic Mo and the regenerated MoSe₂ are slightly different from those of corresponding Mo foil and pristine TS-MoSe₂, which may be caused by the ligand effect of imidazole and amorphous nature, respectively^{41,42}. A similar change trend is also observed in the Se K-edge XANES (Supplementary Fig. 23,24a-d). Specifically, during the initial discharging process, there are two obvious peaks located at 12661.08 and 12668.2 eV in the XANES spectra of TS-MoSe₂, which can be assigned to MoSe₂⁴³. Nevertheless, these two peaks disappear and a new peak appears at 12666.5 V upon discharging to 0.01 V, which corresponds to the generation of the discharged product Na₂Se. During the subsequent charging process, these peaks return to the original state, further indicating that the conversion reaction shows good reversibility. Additionally, the k-space data involved in the manuscript has been provided in Supplementary Fig. 21,22,23,25,26,29. **(Please see page 12 in the revised manuscript, Fig. 4 and Supplementary Fig. 21,22,23,25,26,29.)**

In addition, to further support the conclusion of reversible sodium storage of TS-MoSe₂, we newly supplied *ex-situ* XPS to detect its structural evolution during discharging and charging processes. During the whole evolution processes of discharging and charging, ten voltages were selected to evaluate the structural transformation of the TS-MoSe₂ electrode. As shown in the Mo 3d XPS spectra (Fig. 3a), at the beginning of the discharging process (1.8 and 1.5 V), two main characteristic peaks at 228.83 and 231.93 eV that are related to 3d_{5/2} and 3d_{3/2} of Mo⁴⁺ in MoSe₂ slightly shift towards the low binding energy, indicating the formation of the Na_xMoSe₂ intermediate. With further discharging (1.0 and 0.4 V), a new component with lower binding energies at 227.43 (Mo 3d_{5/2}) and 230.53 eV (Mo 3d_{3/2}) appears and it can be assigned to metallic Mo³⁸, suggesting that the Na_xMoSe₂ has partly transformed into the metallic Mo. At fully discharged state, the Na_xMoSe₂ completely disappears and only metallic Mo is detected. Correspondingly, the Se 3d peak at 54.5 eV first shifts to

higher binding energy, and then restores to the original position, manifesting that Na₂Se finally forms through the polyselenide Na₂(Se)_{1+n} (n > 1) during the discharging process (Fig. 3c)³⁹. Afterwards, in the following charging process, the peaks of both Mo 3d and Se 3d core levels can be fully recovered to their pristine state for TS-MoSe₂, and in contrast, for unstrained MoSe₂, the metallic Mo is always present and meanwhile the elemental Se is eventually generated (Supplementary Fig. 17a,18a). These changes can be observed more visually in corresponding 2D mapping images of the Mo 3d and Se 3d XPS spectra (Fig. 3b and Supplementary Fig. 17b,18b,19), which demonstrate that the strain engineering enables TS-MoSe₂ to follow highly reversible sodium storage mechanism in the discharging and charging processes. **(Please see page 10 in the revised manuscript, Fig. 3 and Supplementary Fig. 17,18,19)**

Revised as follows:

Corresponding revisions on page 12 in the revised manuscript:

In addition, the corresponding wiggle/oscillatory features of the post-edge region of the pristine TS-MoSe₂, fully discharged TS-MoSe₂ (D0.01), fully charged TS-MoSe₂ (C3.0), and Mo foil can also reflect the variation in the local structure of TS-MoSe₂ during the electrochemical process. **(Fig. 4c)**. Upon being fully discharged to 0.01 V, the appearance of the fingerprint feature of Mo foil at 20013.2, 20040.3, and 20084.1 eV supports the formation of metallic Mo¹⁶. In contrast, after the full charging, the aforesaid peaks almost recover to the original state of TS-MoSe₂, while the Mo foil related features disappear, indicating that the discharging and charging processes of TS-MoSe₂ during the initial cycle are nearly fully reversible. It should be noted that the Mo K-edge XANES spectra of the formed metallic Mo and the regenerated MoSe₂ are slightly different from those of corresponding Mo foil and pristine TS-MoSe₂, which may be caused by the ligand effect of imidazole and amorphous nature, respectively

41,42.

Fig. 4. Study on discharging and charging processes based on *ex-situ* XAS. *Ex-situ* Mo K-edge XANES spectra of TS-MoSe₂ during the first **a** discharged and **b** charged states. **c** Mo K-edge XANES spectra of pristine TS-MoSe₂, fully discharged TS-MoSe₂ (D0.01), fully charged TS-MoSe₂ (C3.0), and Mo foil. **d** Evolution of Mo K-edge EXAFS during electrochemical cycling. **e** The intensity evolution of the Mo-Se peak in TS-MoSe₂ (2.11 Å, representing the concentration of TS-MoSe₂) and the Mo-Mo peak in metallic Mo (2.69 Å, representing the concentration of Mo) during electrochemical cycling. Mo K-edge XANES and EXAFS spectra of TS-MoSe₂ **f** and MoSe₂ **g** after the first, second, and fifth cycles.

Supplementary Figure 21. The k-space XANES spectra of TS-MoSe₂ (a), MoSe₂ (b), and TS-MoSe₂ at different discharged states (c-f) for Mo K-edge.

Supplementary Figure 22. The k-space XANES spectra of TS-MoSe₂ at different charged states (a-c) for Mo K-edge.

Supplementary Figure 23. The k-space XANES spectra of TS-MoSe₂ during the first discharge (b-e) and charge (f-h) state for Se K-edge.

Supplementary Figure 25. The k-space XANES spectra of TS-MoSe₂ after the second (a,b) and fifth (c,d) cycles for Mo K-edge.

Supplementary Figure 26. The k-space XANES spectra of TS-MoSe₂ after the second (a,b) and fifth (c,d) cycles for Se K-edge.

Supplementary Figure 29. The k-space XANES spectra of MoSe₂ after the first (a,b), second (c,d), and fifth (e,f) cycles for Mo K-edge.

Corresponding revisions on page 10 in the revised manuscript:

Inspired by the positive influence that the strain engineering has achieved on the redox reaction by the DFT calculations, we first performed *ex-situ* XPS measurements to investigate the effect of the tensile strain on the sodium storage processes. During the whole evolution processes of discharging and charging, ten voltages were selected to evaluate the structural transformation of the TS-MoSe₂ electrode. As shown in the Mo 3d XPS spectra (Fig. 3a), at the beginning of the discharging process (1.8 and 1.5

V), two main characteristic peaks at 228.83 and 231.93 eV that are related to $3d_{5/2}$ and $3d_{3/2}$ of Mo^{4+} in MoSe_2 slightly shift towards the low binding energy, indicating the formation of the Na_xMoSe_2 intermediate. With further discharging (1.0 and 0.4 V), a new component with lower binding energies at 227.43 ($\text{Mo } 3d_{5/2}$) and 230.53 eV ($\text{Mo } 3d_{3/2}$) appears and it can be assigned to metallic Mo^{38} , suggesting that the Na_xMoSe_2 has partly transformed into the metallic Mo. At fully discharged state, the Na_xMoSe_2 completely disappears and only metallic Mo is detected. Correspondingly, the Se 3d peak at 54.5 eV first shifts to higher binding energy, and then restores to the original position, manifesting that Na_2Se finally forms through the polyselenide $\text{Na}_2(\text{Se})_{1+n}$ ($n > 1$) during the discharging process (Fig. 3c)³⁹. Afterwards, in the following charging process, the peaks of both Mo 3d and Se 3d core levels can be fully recovered to their pristine state for TS- MoSe_2 , and in contrast, for unstrained MoSe_2 , the metallic Mo is always present and meanwhile the elemental Se is eventually generated (Supplementary Fig. 17a,18a). These changes can be observed more visually in corresponding 2D mapping images of the Mo 3d and Se 3d XPS spectra (Fig. 3b and Supplementary Fig. 17b,18b,19), which demonstrate that the strain engineering enables TS- MoSe_2 to follow highly reversible sodium storage mechanism in the discharging and charging processes.

Fig. 3. Study on discharging and charging processes based on *ex-situ* XPS spectra. a-c *ex-situ* Mo 3d XPS spectra (a) and corresponding mapping image(b), as well as Se 3d XPS spectra (c) of TS- MoSe_2 during the initial discharging and charging processes.

Supplementary Figure 17. *Ex-situ* Mo 3d XPS spectra of MoSe₂ during the initial discharging and charging processes (a) as well as the corresponding mapping image (b).

Supplementary Figure 18. *Ex-situ* Se 3d XPS spectra (a) and corresponding mapping image (b) of MoSe₂ during the initial discharging and charging processes.

Supplementary Figure 19. *Ex-situ* Se 3d XPS mapping image of TS-MoSe₂ during the initial discharging and charging processes.

Supplementary references:

40. Hu X., Zhu R., Wang B., Wang H. & Liu X. Sn catalyst for efficient reversible conversion between MoSe₂ and Mo/Na₂Se for high-performance energy storage. *Chem. Eng. J.* **440**, 135819 (2022).
41. Karger Z., et al. Selenium and selenium-sulfur cathode materials for high-energy rechargeable magnesium batteries. *J. Power Sources* **323**, 213-219 (2016).

Question 7.

Finally, supposing every claim was well supported, I would not consider this material's potential for actual applications in sodium-ion batteries because it still suffers from the stress and strain issues common to all conversion-type electrodes. Highly desirable research in this field should address this issue first.

Response:

Indeed, as you said, the conversion-type electrodes always suffer from severe stress and strain issues, thus leading to structural failure, which limits their practical applications in sodium-ion batteries. While for MoSe₂, its irreversible sodium storage eventually evolves into a Na-Se battery, which will suffer from the severe shuttle effect of polyselenides and poor structural stability, thus leading to rapid capacity attenuation. This makes it more difficult to clearly distinguish the actual loss of MoSe₂ electrode material from the irreversible reactions or from the stress and strain issues.

Based on this fact, we firstly focused on regulating the sodium storage mechanism of MoSe₂ by constructing the out-plane tensile strain and in-plane compressive strain. The results indicated that during the discharging process, our synthesized strained MoSe₂ can transfer its strain to the discharged product Mo, thus enabling highly activity of Mo and accordingly achieving highly reversible sodium storage of MoSe₂. Meanwhile, we also supplemented the full cell performance of TS-MoSe₂//Na₃V₂(PO₄)₂O₂F in the revised manuscript (**Supplementary Fig. 35**). It can be found that it delivers a performance comparable or superior to many reported full cell performances. (**Please see page 17 in the revised manuscript and Supplementary Fig. 35**)

Thanks for your professional comments once again. According to your valuable suggestions, we will strive to address stress and strain issues based on the reversible sodium storage of MoSe₂ to further improve its electrochemical performance in our future research.

Revised as follows:

Corresponding revisions on page 17 in the revised manuscript:

In view of the superior sodium storage performance of TS-MoSe₂ in the half cells, the Na-ion full cells were further assembled with homemade Na₃V₂(PO₄)₂O₂F (NVPOF) as a cathode to preliminarily assess its practicability as an anode for SIBs (**Supplementary Fig. 34,35**). As shown in **Supplementary Fig. 35b,c**, the full cells

exhibit good cycling performance and the capacity can still maintain 434.9 mA h g⁻¹ after 200 cycles at 0.2 A g⁻¹ (based on the mass of the anode). The full cells also present superior rate capabilities (**Supplementary Fig. 35d**), in which about 70.2% of the capacity can be retained even when the current density increases by 50-folds from 0.1 to 5 A g⁻¹. The good rate capabilities endow the full cells with a specific energy of 108.6 Wh kg⁻¹ at a power density of 19.0 W kg⁻¹, and even 74.1 Wh kg⁻¹ at a power density of 648.5 W kg⁻¹ (based on the total mass of the electrode materials), which are comparable or superior to those of many reported full cells (**Supplementary Fig. 35e**).

Supplementary Figure 35. Electrochemical performances of the TS-MoSe₂/NVPOF full cell. (a) Schematic illustration of the TS-MoSe₂/NVPOF full cell. (b) Cycling performance at 0.2 A g⁻¹ and (c) corresponding galvanostatic charge-discharge curves. (d) Rate performance. (e) Ragone plots of gravimetric energy density vs. power density (**Ref-3**: SnP₂O₇/rGO//Na₃V₂(PO₄)₃/C; **Ref-4**: MoSe₂/C//NVP/C; **Ref-5**: graphite/prussian white; **Ref-6**: Na₂Fe₂(SO₄)₃/Ti₂CT_x; **Ref-7**: CoSe₂@NC//NVPOF; **Ref-8**: NTP@rGO//Na₃V₂(PO₄)₃/C; **Ref-9**: CNT//Na₃V₂(PO₄)₃, **Ref-10**: NaTi₂(PO₄)₃/Na₂PB; **Ref-11**: Na-ion V₂O₅-CNT//activated carbon; **Ref-12**: graphite//Na_{0.7}CoO₂; **Ref-13**: NaTi₂(PO₄)₃/Na_{0.44}MnO₂; **Ref-14**: MoSe₂/N,P-rGO//Na₃V₂(PO₄)₃/C).

Reviewer #3:

There are several published works on the topic of transition metal dichalcogenides for rechargeable alkali metal ion batteries. These materials tend to suffer from high first cycle loss, voltage hysteresis, considerable volume changes leading to capacity loss etc. In addition, high cost of such materials is a major barrier for use in low-cost sodium ion batteries. In the present work authors describe synthesis of a MoSe₂ type TMD, which under localized tensile deformation shows improved electrochemical performance as battery electrode over neat/pristine MoSe₂. The work seem to be interesting but has some major flaws that need to be rectified before publication in any scientific journal.

Thank you very much for your constructive comments. These professional comments and suggestions are very helpful for us to improve the quality of our manuscript. We have revised the manuscript in accordance with your suggestions. Meanwhile, we also supplemented the full cell performance of TS-MoSe₂//Na₃V₂(PO₄)₂O₂F in the revised manuscript (**Supplementary Fig. 35**). It can be found that it delivers a performance comparable or superior to many reported full cell performances. Thanks for your valuable suggestions once again. Our responses to your questions are below:

Supplementary Figure 35. Electrochemical performances of the TS-MoSe₂/NVPOF full cell. (a) Schematic illustration of the TS-MoSe₂/NVPOF full cell. (b) Cycling performance at 0.2 A g⁻¹ and (c) corresponding galvanostatic charge-discharge curves. (d) Rate performance. (e) Ragone plots of gravimetric energy density vs. power density (**Ref-3:** SnP₂O₇/rGO//Na₃V₂(PO₄)₃/C; **Ref-4:** MoSe₂/C//NVP/C; **Ref-5:** graphite//prussian white; **Ref-6:** Na₂Fe₂(SO₄)₃//Ti₂CT_x; **Ref-7:** CoSe₂@NC//NVPOF; **Ref-8:** NTP@rGO//Na₃V₂(PO₄)₃/C; **Ref-9:** CNT//Na₃V₂(PO₄)₃, **Ref-10:** NaTi₂(PO₄)₃//Na₂PB; **Ref-11:** Na-ion V₂O₅-CNT//activated carbon; **Ref-12:** graphite//Na_{0.7}CoO₂; **Ref-13:** NaTi₂(PO₄)₃//Na_{0.44}MnO₂; **Ref-14:** MoSe₂/N,P-rGO//Na₃V₂(PO₄)₃/C).

Question 1.

Fig 2f: what is the additional peak observed in TS-MoSe₂ close to 325 cm⁻¹? same for Figures g and j.

Response:

Thanks for your kind suggestions. The peak close to 325 cm⁻¹, corresponding to the B_{2g}¹ of MoSe₂, has been marked in the revised manuscript. Additionally, the corresponding Mo-Mo bond in the second coordination shell for MoSe₂ and MoN has also been supplied in Figs. 2g and j. **(Please see page 7 in the revised manuscript)**

Revised as follows:

Fig. 2. Materials synthesis and characterizations. **f** Raman spectra and **g** The normalized Mo K-edge EXAFS spectra (circle) of TS-MoSe₂ and MoSe₂ as well as the corresponding EXAFS fitting curves (line). **j** The normalized Mo K-edge EXAFS spectra of MoN and the fully discharged product of TS-MoSe₂ (TS-MoSe₂-D0.01) and MoSe₂ (MoSe₂-D0.01).

Question 2.

Fig 3b: The in-situ Raman is inconclusive. It is not uncommon to observe such subtle changes with-in the same sample from one location to another. Several different samples need to be tested to confirm the conclusions related to shift/suppression/reemergence of E_{2g}¹ band.

Response:

We are grateful for your kind and valuable suggestions. Indeed, in some cases, the Raman signal may be weak, especially for amorphous electrochemical charged/discharged products with increased disorder. Thus, the mechanism analysis only based on *in-situ* Raman is inconclusive. Following your professional suggestions, we repeated the *in-situ* Raman test under the same conditions and the results still show a similar pattern as before. **(Please see page 11 in the revised manuscript and Supplementary Fig. 20)**

More importantly, we further provided *ex-situ* XPS to support the conclusion of reversible sodium storage of TS-MoSe₂. Specifically, during the whole evolution processes of discharging and charging, ten voltages were selected to evaluate the

structural transformation of the TS-MoSe₂ electrode. As shown in the Mo 3d XPS spectra (Fig. 3a), at the beginning of the discharging process (1.8 and 1.5 V), two main characteristic peaks at 228.83 and 231.93 eV that are related to 3d_{5/2} and 3d_{3/2} of Mo⁴⁺ in MoSe₂ slightly shift towards the low binding energy, indicating the formation of the Na_xMoSe₂ intermediate. With further discharging (1.0 and 0.4 V), a new component with lower binding energies at 227.43 (Mo 3d_{5/2}) and 230.53 eV (Mo 3d_{3/2}) appears and it can be assigned to metallic Mo³⁸, suggesting that the Na_xMoSe₂ has partly transformed into the metallic Mo. At fully discharged state, the Na_xMoSe₂ completely disappears and only metallic Mo is detected. Correspondingly, the Se 3d peak at 54.5 eV first shifts to higher binding energy, and then restores to the original position, manifesting that Na₂Se finally forms through the polyselenide Na₂(Se)_{1+n} (n > 1) during the discharging process (Fig. 3c)³⁹. Afterwards, in the following charging process, the peaks of both Mo 3d and Se 3d core levels can be fully recovered to their pristine state for TS-MoSe₂, and in contrast, for unstrained MoSe₂, the metallic Mo is always present and meanwhile the elemental Se is eventually generated (Supplementary Fig. 17a,18a). These changes can be observed more visually in corresponding 2D mapping images of the Mo 3d and Se 3d XPS spectra (Fig. 3b and Supplementary Fig. 17b,18b,19), which demonstrate that the strain engineering enables TS-MoSe₂ to follow highly reversible sodium storage mechanism in the discharging and charging processes. **(Please see pages 10 and 22 in the revised manuscript, Fig. 3 and Supplementary Fig. 17, 18, 19)**

Revised as follows:

Corresponding revisions on page 11 in the revised manuscript:

Furthermore, to exclude the influence of testing errors, we repeated the *in-situ* Raman testing and the experimental results are basically consistent (**Supplementary Fig. 20**).

Supplementary Figure 20. *In-situ* Raman spectra of TS-MoSe₂ during the initial discharging and charging processes (the experimental results from the repeated test of *in-situ* Raman).

Corresponding revisions on page 10 in the revised manuscript:

During the whole evolution processes of discharging and charging, ten voltages were selected to evaluate the structural transformation of the TS-MoSe₂ electrode. As shown in the Mo 3d XPS spectra (**Fig. 3a**), at the beginning of the discharging process (1.8 and 1.5 V), two main characteristic peaks at 228.83 and 231.93 eV that are related to 3d_{5/2} and 3d_{3/2} of Mo⁴⁺ in MoSe₂ slightly shift towards the low binding energy, indicating the formation of the Na_xMoSe₂ intermediate. With further discharging (1.0 and 0.4 V), a new component with lower binding energies at 227.43 (Mo 3d_{5/2}) and 230.53 eV (Mo 3d_{3/2}) appears and it can be assigned to metallic Mo³⁸, suggesting that the Na_xMoSe₂ has partly transformed into the metallic Mo. At fully discharged state, the Na_xMoSe₂ completely disappears and only metallic Mo is detected. Correspondingly, the Se 3d peak at 54.5 eV first shifts to higher binding energy, and then restores to the original position, manifesting that Na₂Se finally forms through the polyselenide Na₂(Se)_{1+n} (n > 1) during the discharging process (**Fig. 3c**)³⁹. Afterwards, in the following charging process, the peaks of both Mo 3d and Se 3d core levels can be fully recovered to their pristine state for TS-MoSe₂, and in contrast, for unstrained MoSe₂, the metallic Mo is always present and meanwhile the elemental Se is eventually generated (**Supplementary Fig. 17a,18a**). These changes can be observed more visually in corresponding 2D mapping images of the Mo 3d and Se 3d XPS spectra (**Fig. 3b and Supplementary Fig. 17b,18b,19**), which demonstrate that the strain engineering enables TS-MoSe₂ to follow highly reversible sodium storage mechanism in the discharging and charging processes.

Fig. 3. Study on discharging and charging processes based on *ex-situ* XPS spectra. **a-c** *ex-situ* Mo 3d XPS spectra (a) and corresponding mapping image (b), as well as Se 3d XPS spectra (c) of TS-MoSe₂ during the initial discharging and charging processes.

Supplementary Figure 17. *Ex-situ* Mo 3d XPS spectra of MoSe₂ during the initial discharging and charging processes (a) as well as the corresponding mapping image (b).

Supplementary Figure 18. *Ex-situ* Se 3d XPS spectra (a) and corresponding mapping image (b) of MoSe₂ during the initial discharging and charging processes.

Supplementary Figure 19. *Ex-situ* Se 3d XPS mapping image of TS-MoSe₂ during the initial discharging and charging processes.

Corresponding revisions on page 22 in the revised manuscript:

***Ex-situ* XPS characterizations:** firstly, the battery was discharged and charged up to the required potential using a LAND workstation at a current density of 0.02 A g⁻¹. Then, the battery was disassembled in a glove box to collect the electrode sheet. Afterwards, the resulting electrode sheet was washed with dimethyl carbonate (DMC) to remove any residual salts. Finally, the tested electrode sheet was transported with a vacuum-transfer-module from the glove box to the XPS test system to avoid component changes when exposed to air. Furthermore, the electrode sheet was also etched with Ar⁺

ion beam before the test to further avoid interference from surface SEI.

Question 3.

Fig.4: Caption reads sodium storage mechanisms--please explain what new mechanisms or electrochemical reactions are observed TS-MoSe₂?

Response:

We are sorry that our imprecise captions in Fig. 3 and 4 may have caused you some confusion. Actually, Fig. 3 and 4 mainly demonstrate that strain engineering enables TS-MoSe₂ to achieve a reversible sodium storage mechanism during the charging and discharging processes rather than new mechanisms, and we have corrected the inappropriate captions and the corresponding subtitle in the revised manuscript. **(Please see pages 11 and 13 the revised manuscript)**

Question 4.

Fig 5. Shows improved gravimetric capacity and rate capability of TS-MoSe₂ over MoSe₂. What is the mass of each electrode? Please supply areal capacity values. Also, mention how many specimens were tested for each type? In addition, without the 1st and 2nd cycle voltage profiles it is not possible to make any conclusions about superiority of TS-MoSe₂ over MoSe₂.

Response:

We are grateful for your kind and valuable suggestions. We have marked the electrode mass involved in each electrochemical performance test in Fig. 5 and provided the corresponding areal capacity curves in Supplementary Fig. 32.

Additionally, we are so sorry that the description of electrochemical tests in the manuscript is incomplete. The number of specimens has been given in the experimental section, that is, 3 specimens were tested for each type of battery performance evaluation to ensure that they have almost identical results. Also, the galvanostatic charge-discharge curves of MoSe₂ have been added in Supplementary Fig. 31. **(Please see page 23 in the revised manuscript, Fig. 5 and Supplementary Fig. 31,32)**

Revised as follows:

Corresponding revisions on page 23 in the revised manuscript:

In order to effectively avoid errors introduced during the testing process, 3 specimens were tested for each type of battery performance evaluation to ensure that they have almost identical results.

Fig. 5. Electrochemical performance. **a** CV curves of TS-MoSe₂ between 0.01 and 3.0 V at a potential sweep speed of 0.1 mV s⁻¹. **b** dQ/dV plots of TS-MoSe₂ and MoSe₂. **c** Cycling and **d** rate performances of TS-MoSe₂ and MoSe₂. **e** Comparison of the rate capacities of TS-MoSe₂ with a series of reported MoSe₂-based anodes. **f** Cycling performances TS-MoSe₂ and MoSe₂ at different temperatures from 50 to -30 °C. **g** Cycling performances of TS-MoSe₂ at -10, -30 °C and MoSe₂ at -30 °C.

Supplementary Figure 31. Galvanostatic charge-discharge curves of (b) MoSe₂ at the current density of 0.1 A g⁻¹ for the 1st, 2nd, 3rd, 50th, and 100th cycles.

Supplementary Figure 32. The areal capacities of TS-MoSe₂ and MoSe₂ electrodes at the current density of 0.1 A g⁻¹.

Question 5.

There is barely any difference in the Na⁺ diffusion coefficient in TS-MoSe₂ over MoSe₂ in Fig 6f. The unit has a factor of 10⁻⁹. In certain regions [voltage values], the calculated values of the diffusion coefficient overlap.

Response:

We appreciate the reviewer very much to notice this detail. Actually, there are still some differences in the Na⁺ diffusion coefficients, and especially in the low de/intercalation states, the calculated D_{Na} values of TS-MoSe₂ are larger than those of MoSe₂, although in some regions their D_{Na} values almost overlap. This may be related to the different sodium storage processes that lead to varied intermediate phases. Specifically, TS-MoSe₂ experiences the in-/de-tercalation and conversion reactions (generally the former has a higher D_{Na} due to weaker interlayer van der Waals forces^{61,62}), while for MoSe₂, Se/Na₂Se becomes the sole redox couple after the initial cycling that only occurs the conversion reaction ($\text{Se} + 2\text{Na}^+ + 2\text{e}^- \leftrightarrow \text{Na}_2\text{Se}$). Thus, the conversion process of the two cases involves similar intermediate phases, thereby resulting in almost same D_{Na} values. **(Please see page 20 in the revised manuscript)**

Revised as follows:

Corresponding revisions on page 20 in the revised manuscript:

Clearly, the calculated D_{Na} values of TS-MoSe₂ are larger than those of MoSe₂ at most of the discharging/charging states, while in some regions their D_{Na} values almost overlap. Based on the foregoing analyses, TS-MoSe₂ experiences the in-/de-tercalation and conversion reactions (generally the former has a higher D_{Na} due to weaker interlayer van der Waals forces^{61,62}), while for MoSe₂, Se/Na₂Se becomes the sole redox couple after the initial cycling that only occurs the conversion reaction ($\text{Se} + 2\text{Na}^+ + 2\text{e}^- \leftrightarrow \text{Na}_2\text{Se}$). Thus, the conversion process of the two cases involves similar intermediate phases, thereby resulting in almost same D_{Na} values.

Supplementary references:

61. Chen B., et al. Transition metal dichalcogenides for alkali metal ion batteries: engineering strategies at the atomic level. *Energy Environ. Sci.* **13**, 1096–1131 (2020).
62. Xu J., Zhang J., Zhang W. & Lee C. S. Interlayer nanoarchitectonics of two-dimensional transition-metal dichalcogenides nanosheets for energy storage and conversion applications. *Adv. Energy Mater.* **7**, 1700571 (2017).

Question 6.

Very sadly, the authors failed to cite the most important initial works on chalcogenides for Na⁺ batteries [first report on TMD for Na⁺ battery by Singh: <https://pubs.acs.org/doi/abs/10.1021/nn406156b>] and sustainable Na⁺ batteries [Nodal review article by Larcher/Tarascon on Na⁺ batteries in Nature Chemistry: <https://www.nature.com/articles/nchem.2085>].

Response:

We greatly appreciate your constructive comments. The groundbreaking research of Singh on the application of TMD to Na⁺ battery provides an alternative to the development on anode materials for sodium ion battery and the review article for green and sustainable Na⁺ batteries by Tarascon and Larcher points the way for the subsequent development of Na⁺ batteries, which have been cited and updated as ref. 8 and 4 in references. **(Please see ref. 4 and 8 in the revised manuscript)**

Question 7.

Overall, this is an interesting work with a fancy title [please consider removing terms like "strained-gene" from the title] that requires some additional experimental validation to support the conclusions about the superiority of TS-MoSe₂ over MoSe₂. The work is obviously of some fundamental nature but of little use from practicality of TMD-based materials in batteries.

Response:

Thank you very much for your kind and professional suggestions. We have removed the terms of "strained-gene" from the title. In addition, we also newly added *ex-situ* XPS to further support the conclusion of reversible sodium storage of TS-MoSe₂. Specifically, during the whole evolution processes of discharging and charging, ten voltages were selected to evaluate the structural transformation of the TS-MoSe₂ electrode. As shown in the Mo 3d XPS spectra (Fig. 3a), at the beginning of the discharging process (1.8 and 1.5 V), two main characteristic peaks at 228.83 and 231.93 eV that are related to 3d_{5/2} and 3d_{3/2} of Mo⁴⁺ in MoSe₂ slightly shift towards the low binding energy, indicating the formation of the Na_xMoSe₂ intermediate. With further discharging (1.0 and 0.4 V), a new component with lower binding energies at 227.43 (Mo 3d_{5/2}) and 230.53 eV (Mo 3d_{3/2}) appears and it can be assigned to metallic Mo³⁸, suggesting that the Na_xMoSe₂ has partly transformed into the metallic Mo. At fully discharged state, the Na_xMoSe₂ completely disappears and only metallic Mo is detected. Correspondingly, the Se 3d peak at 54.5 eV first shifts to higher binding energy, and

then restores to the original position, manifesting that Na₂Se finally forms through the polyselenide Na₂(Se)_{1+n} (n > 1) during the discharging process (Fig. 3c)³⁹. Afterwards, in the following charging process, the peaks of both Mo 3d and Se 3d core levels can be fully recovered to their pristine state for TS-MoSe₂, and in contrast, for unstrained MoSe₂, the metallic Mo is always present and meanwhile the elemental Se is eventually generated (Supplementary Fig. 17a,18a). These changes can be observed more visually in corresponding 2D mapping images of the Mo 3d and Se 3d XPS spectra (Fig. 3b and Supplementary Fig. 17b,18b,19), which demonstrate that the strain engineering enables TS-MoSe₂ to follow highly reversible sodium storage mechanism in the discharging and charging processes. **(Please see page 10 in the revised manuscript, Fig. 3 and Supplementary Fig. 17, 18, 19)**

Revised as follows:

Corresponding revisions on page 10 in the revised manuscript:

During the whole evolution processes of discharging and charging, ten voltages were selected to evaluate the structural transformation of the TS-MoSe₂ electrode. As shown in the Mo 3d XPS spectra (**Fig. 3a**), at the beginning of the discharging process (1.8 and 1.5 V), two main characteristic peaks at 228.83 and 231.93 eV that are related to 3d_{5/2} and 3d_{3/2} of Mo⁴⁺ in MoSe₂ slightly shift towards the low binding energy, indicating the formation of the Na_xMoSe₂ intermediate. With further discharging (1.0 and 0.4 V), a new component with lower binding energies at 227.43 (Mo 3d_{5/2}) and 230.53 eV (Mo 3d_{3/2}) appears and it can be assigned to metallic Mo³⁸, suggesting that the Na_xMoSe₂ has partly transformed into the metallic Mo. At fully discharged state, the Na_xMoSe₂ completely disappears and only metallic Mo is detected. Correspondingly, the Se 3d peak at 54.5 eV first shifts to higher binding energy, and then restores to the original position, manifesting that Na₂Se finally forms through the polyselenide Na₂(Se)_{1+n} (n > 1) during the discharging process (**Fig. 3c**)³⁹. Afterwards, in the following charging process, the peaks of both Mo 3d and Se 3d core levels can be fully recovered to their pristine state for TS-MoSe₂, and in contrast, for unstrained MoSe₂, the metallic Mo is always present and meanwhile the elemental Se is eventually generated (**Supplementary Fig. 17a,18a**). These changes can be observed more visually in corresponding 2D mapping images of the Mo 3d and Se 3d XPS spectra (**Fig. 3b and Supplementary Fig. 17b,18b,19**), which demonstrate that the strain engineering enables TS-MoSe₂ to follow highly reversible sodium storage mechanism in the discharging and charging processes.

Fig. 3. Study on discharging and charging processes based on *ex-situ* XPS spectra. **a-c** *ex-situ* Mo 3d XPS spectra (a) and corresponding mapping image (b), as well as Se 3d XPS spectra (c) of TS-MoSe₂ during the initial discharging and charging processes.

Supplementary Figure 17. *Ex-situ* Mo 3d XPS spectra of MoSe₂ during the initial discharging and charging processes (a) as well as the corresponding mapping image (b).

Supplementary Figure 18. *Ex-situ* Se 3d XPS spectra (a) and corresponding mapping image (b) of MoSe₂ during the initial discharging and charging processes.

Supplementary Figure 19. *Ex-situ* Se 3d XPS mapping image of TS-MoSe₂ during the initial discharging and charging processes.

REVIEWERS' COMMENTS

Reviewer #1 (Remarks to the Author):

The authors addressed all the comments given by the reviewers. Lots of additional characterizations were undertaken, that provided solid support to the discussions. The quality of the revised manuscript was largely improved. There are some typos in the revised manuscript, and the authors need to carefully check them. In all, I think the present manuscript could be accepted to be published,

Reviewer #3 (Remarks to the Author):

Revised version is much improved. Comments have been addressed to satisfaction.

Response to Reviewers' comments:

Reviewer #1:

The authors addressed all the comments given by the reviewers. Lots of additional characterizations were undertaken, that provided solid support to the discussions. The quality of the revised manuscript was largely improved. There are some typos in the revised manuscript, and the authors need to carefully check them. In all, I think the present manuscript could be accepted to be published.

Response:

Thank you very much for your valuable comments. We have corrected the mistakes and carefully checked the whole manuscript.

Reviewer #3:

Revised version is much improved. Comments have been addressed to satisfaction.

Response:

We greatly appreciate your kind comments.